



# Use of Large-Eddy simulations to design an adaptive sampling strategy to assess cumulus cloud heterogeneities by Remotely Piloted Aircraft

Nicolas Maury[1], Gregory C. Roberts[1,2], Fleur Couvreux[1], Titouan Verdu[3,4], Pierre Narvor[3], Najda Villefranque[1], Simon Lacroix[3], and Gautier Hattenberger[4]

[1]Centre National de Recherches Météorologiques, Université de Toulouse, Météo-France, CNRS, Toulouse, France
[2]Scripps Institution of Oceanography, University of California San Diego, La Jolla, USA
[3]Laboratoire d'Analyse et d'Architecture des Systèmes, Université de Toulouse, CNRS, Toulouse, France
[4]Ecole Nationale de l'Aviation Civile, Université de Toulouse, Toulouse, France

**Correspondence:** Nicolas Maury (nicolas.maury@meteo.fr) and Gregory C. Roberts (greg.roberts@meteo.fr)

**Abstract.**

Trade wind cumulus clouds have a significant impact on the earth's radiative balance, due to their ubiquitous presence and significant coverage in subtropical regions. Many numerical studies and field campaigns have focused
on better understanding the thermodynamic and macroscopic properties of cumulus clouds with ground-based and satellite remote sensing as well as in-situ observations. Aircraft flights have provided a significant contribution, but their resolution remains limited by rectilinear transects and fragmented temporal data of individual clouds. To provide a higher spatial and temporal resolution, Remotely Piloted Aircraft (RPA) can now be employed for direct observations, using numerous technological advances, to map the microphysical cloud structure and to study
entrainment mixing. In fact, the numerical representation of mixing processes between a cloud and the surrounding air has been a key issue in model parameterizations for decades. To better study these mixing processes as well as their impacts on cloud microphysical properties, the manuscript aims to improve exploration strategies that can be implemented by a fleet of RPAs.

Here, we use a Large-Eddy simulation (LES) of oceanic cumulus clouds to design adaptive sampling strategies. An
implementation of the RPA flight simulator within high-frequency LES outputs (every 5 s) allows to track individual clouds. A Rosette sampling strategy is used to explore clouds of different sizes, static in time and space. The adaptive sampling carried out by these explorations is optimized using one or two RPAs and with or without Gaussian Process Regression (GPR) mapping, by comparing the results obtained with those of a reference simulation, in particular the total liquid water content (LWC) and the LWC distributions in a horizontal cross section. Also, a sensitivity test of
lengthscale for GPR mapping is performed. The results of exploring a static cloud are then extended to a dynamic case of a cloud evolving with time, to assess the application of this exploration strategy to study the evolution of cloud heterogeneities.





## 1  Introduction

Cumulus clouds are ubiquitous over the subtropical oceanic regions and cover more than 20 % of the oceanic surface
on annual average (Eastman et al., 2011). They mainly interact with the shortwave radiation and therefore exert a
net cooling effect on the Earth system. They also modulate the water and energy cycles of the atmosphere through
vertical transfer from the sub-cloud layer to the cloud layer. Cumulus clouds are therefore a key element of the
climate system (Park et al., 2017). Their representation in Global Circulation Models (GCMs) has been shown to be
responsible for large uncertainties in the climatic response (Andrews et al., 2012). Due to their grid scales between
10 to 100 kilometers, GCMs cannot explicitly represent shallow clouds, and use parameterizations to represent the
impacts of such clouds on the climate response. One of the biggest uncertainties in understanding the impacts of
cumulus clouds on the water and energy cycle is related to mixing processes (Sanchez et al., 2020). Mixing processes
impact cloud microphysical properties by creating heterogeneities of thermodynamical variables, diluting the liquid
water content and reducing the cloud albedo. The studies on these processes often rely on the analysis of Large-Eddy
Simulations (LES) that reproduce average properties of shallow convection (Guichard and Couvreux, 2017, Siebesma
and Jonker, 2000, Neggers et al., 2003, Heus and Jonker, 2008). However such models, with a horizontal resolution
of a few tens of meters, still use parameterizations to represent cloud microphysics and small-scale turbulence to
correctly reproduce sub-grid heterogeneities inside cumulus clouds such as sub-grid scale liquid water content (LWC)
variability resulting from mixing processes at the cloud-air interface.

Observations of cumulus horizontal structures in the western Atlantic Ocean have been obtained from field cam-
paigns such as BOMEX (Barbados Oceanographic and Meteorological EXperiment, Holland and Rasmusson, 1973),
SCMS (Burnet and Brenguier, 2007), CARRIBA (Hoffmann et al., 2014), RICO (Rauber et al., 2007), and cloud
instrumentation continues to improve (i.e. the Fast-FSSP (Brenguier et al., 1998) to the HOLODEC (Holographic
Detector for Clouds; Fugal and Shaw, 2009)). However, the sampling strategy that relies on one or two transects
through the same cloud only is not sufficient to reconstruct the cloud cross section or follow its evolution. Aircraft
transects have been shown to induce a bias in the sampling of clouds (Mallaun et al., 2019) by oversampling the
cloud core. Some measurement field campaigns have allowed a re-sampling in clouds with aircraft (Burnet and Bren-
guier, 2007) and with sensors suspended below a helicopter during the CARRIBA campaign (Siebert et al., 2006,
Katzwinkel et al., 2014).

These campaigns serve as a basis for the construction of well-established case studies on which LES have been
used to develop and evaluate shallow cloud parameterization (Siebesma et al., 2003, vanZanten et al., 2011). These
LES reproduce the cloud field and allow the study of isolated clouds in details, notably at high spatial and temporal
resolution (Zhao and Austin, 2005a). Nonetheless, LES also rely on parameterizations, especially for the sub-grid
turbulence and the cloud microphysics including cloud droplet number concentrations within the grid.

Over the past two decades, Remotely Piloted Aircrafts (RPAs) have emerged as a viable tool for observing aerosols
and clouds (Roberts et al., 2008, Sanchez et al., 2017, Calmer et al., 2019). Their ability to operate as a fleet and follow





complex trajectories based on adaptive sampling are an asset which allows a detailed comparison with high-resolution simulations. Previous studies have developed tools to implement RPAs in LES to optimize trajectories within the cloud environment with the objective to maximize information gain while minimizing energy consumption (Reymann et al., 2018). In this study, we focus on obtaining relevant meteorological data to observe cloud heterogeneities and mixing. A powerful tool in RPA cloud tracking is Gaussian Process Regression (GPR) mapping during flights to best guide the RPAs pattern and during post-processing to reconstruct cumulus field (Renzaglia et al., 2016).

The objective of this study is to simulate RPA flights in LES output in order to optimize an adaptive sampling strategy to provide sufficient microphysic and thermodynamic information within a marine cumulus cloud to quantify the mixing processes. This study is part of the NEPHELAE project (Network for studying Entrainment and microPHysics of cLouds using Adaptive Exploration), which aims to design and develop an automated fleet of RPAs to track a cloud from the beginning to the end of its life cycle. Section 2 presents the LES model, cloud identification methods, as well as the details of the RPA flight parameters. Section 3 highlights the results of the LES case study with an overview of the cumulus field. We first classify individual simulated clouds into three categories based on their volume. We then select one cloud representative of each category and analyze the evolution of their macrophysical and thermodynamical properties, by comparing the exploration strategy and the capacity of the RPAs to reconstruct the microphysical and macrophysical fields for static and dynamic cases. This study highlights benefits of adaptive sampling and GPR mapping and illustrates the potential of RPAs to address long-standing challenges in observing clouds.

## 2 Methodology

### 2.1 Cloud simulation

#### 2.1.1 BOMEX: a case of marine convection

The numerical simulations focus on the period between 22 to 23 June 1969 of the Phase 3 of the BOMEX campaign characterized by the presence of a strong inversion at the top of the boundary layer (Siebesma and Cuijpers, 1995). This case has been chosen because it represents a typical undisturbed non-precipitating trade cumulus cloud field.

The BOMEX case was the subject of a model intercomparison exercise (Siebesma et al., 2003) with 10 LESs based on different models. The LES simulations all start with the same initial profiles of total mixing water ratio ($q_t$) and liquid potential temperature ($\theta_l$) from sea-level to boundary layer top measured by radiosondes. These LES models use prescribed constant surface latent and sensible heat fluxes ($8 \times 10^3 \, \mathrm{K \, m \, s^{-1}}$, $5.2 \times 10^{-5} \, \mathrm{K \, m \, s^{-1}}$) and prescribed large-scale and radiative forcing (Siebesma and Cuijpers, 1995). Total cloud cover and liquid water path were well-represented (Fig.1). These LES also correctly reproduced the observed vertical thermodynamical structure and turbulent fluxes for this period (Nitta and Esbensen, 1974). The horizontal winds is initialized with



U= $8.75\,\mathrm{m\,s^{-1}}$ and V= $0\,\mathrm{m\,s^{-1}}$ between sea level and 700 m.asl (meters above sea level) and decreases linearly until U=$-4.61\,\mathrm{m\,s^{-1}}$ at 3000 m.asl.

### 2.1.2 Meso-NH model and configuration

Meso-NH, a French non-hydrostatic mesoscale atmospheric model (Lac et al., 2018) is used in LES-mode to simulate the BOMEX case and the results are compared to the LES intercomparison of Siebesma et al. (2003) in Sect. 3. The thermodynamical variables are advected with the piecewise parabolic model, while the wind is advected with a fourth-order centered scheme coupled to an explicit fourth-order centered Runge-Kutta time splitting (Lunet et al., 2017). Lateral boundary conditions are cyclic and a damping layer is applied at the top of the domain to prevent the reflection of gravity waves. The three-dimensional turbulence scheme from Cuxart et al. (2000) is based on a prognostic equation for the sub-grid turbulence kinetic energy with a Deardorff mixing length (Deardorff, 1980). Trade cumuli contain only liquid water, and therefore, only a warm bulk one-moment microphysic scheme is used. Long-wave radiative cooling, corresponding to the effect of clear-sky emissions, is prescribed for each atmospheric column as a temperature tendency. An "all-or-nothing" grid scheme is used so the grid is either entirely saturated (cloud) or entirely clear (no cloud).

The BOMEX case (Siebesma et al., 2003) was re-simulated for this study using Meso-NH LES with $\Delta x=\Delta y=\Delta z=25$ m - a higher horizontal resolution than used for the intercomparison study ($\Delta x=\Delta y= 100$ m, $\Delta z=40$ m, (Siebesma and Cuijpers, 1995, Siebesma et al., 2003)). The Meso-NH LES was conducted on two different horizontal domains: the same domain as the intercomparison study (6.4 km × 6.4 km × 4 km with 256 × 256 × 160 meshes) named 6.4km_MNH and a domain four-times the surface area (12.8 km × 12.8 km × 4 km with 512 × 512 × 160 meshes) named 12.8km_MNH. The duration of these simulations is six hours where the first two hours of the simulation are discarded as spin-up. The 12.8km_MNH run continued for 30 minutes longer during which 3D fields were outputted every 5 seconds in order to have high temporal resolution of cloud fields, named HFS for High Frequency Simulation.

### 2.1.3 LES validation

To validate the high resolution Meso-NH, the total cloud cover (TCC) is compared to results from the reference intercomparison study (Siebesma et al., 2003) as shown in Fig. 1. The TCC and liquid water path (LWP) stabilize after the spin-up to $\sim 15\,\%$ and $\sim 5\,\mathrm{g\,m^{-2}}$, respectively. From the 2nd to the 6th hour, the TCC of both Meso-NH simulations remains within the standard deviation of the intercomparison study (Siebesma et al., 2003) with more fluctuations for the 6.4 km domain.

At the end of the simulation, TCC from 6.4km_MNH is slightly higher (+ 5 %) than reported in Siebesma et al., 2003, while LWP remains nearly the same.

The vertical profiles of turbulent flux of $q_t$, $\theta_l$, wind, turbulent kinetic energy (TKE) and LWC are also near the mean and within the variability of the intercomparison ensemble presented in Siebesma et al., 2003 (not shown). The TCC in HFS (shaded area in Fig. 1) varies during the 30 minutes between 11.9 % and 15.3 %, while the mean LWP



in the domain is between $4.30$ and $6.07\,\mathrm{g\,m^{-2}}$. In the following sections, the analysis focuses on the high temporal frequency outputs (HFS) in order to study the life cycle of individual clouds.

### 2.1.4 Cloud identification method

One of the main objectives is to be able to characterize a cloud entire life cycle, including the formation phase

when updrafts dominate and the dissipation phase when downdrafts dominate. In order to do that, we need to track individual clouds as the function of time while exploiting the high spatial and temporal resolution of the LES.

We first define clouds as coherent 3D structures made of at least 8 contiguous cells containing a LWC $>$ $1 \times 10^{-3}\,\mathrm{g\,kg^{-1}}$ and overlapping at least two vertical levels, i.e clouds thinner than 50 m or smaller than $1.25 \times 10^{-4}\,\mathrm{km^3}$ are filtered out. In order to follow individual clouds, we apply a method of cloud identification (Brient et al., 2019)

as a function of time, t, for the first time. As shown in Fig. 2, the cloud identification method uses matrices of contiguity that isolates a cell and defines it as belonging to cloud N. For each cloudy cell, the method identifies the neighboring cells (per face, edge or corner). If one of them is already tagged as a cloudy cell, it will get the same tag (Fig. 2, $t_0$). This method also uses contiguity in time, with the criterion that a face, an edge or a corner of a cloudy cell at $t_n$ touches a cloudy cell at $t_{n-1}$ (Fig. 2, $t_1$). However, the advection of the cloud must be within a spatial limit

between two time steps defined by the Courant-Friedrichs-Lewy condition (CFL = $(U\Delta t/\Delta x) \leq 1$). If the cloud moves two or more lengths during one time step, the cloud identification method can lead to errors. In this study, the advection wind U is between $5\,\mathrm{m\,s^{-1}}$ and $8\,\mathrm{m\,s^{-1}}$, the horizontal resolution is $\Delta x$=25 m, and outputs every are 5 seconds, which yields a CFL between 1 and 1.6 and does not meet the CFL condition. To solve this issue, only cumuli with an overall dimension at least 3 times larger than a mesh (cloud width $\geq$ 75 m) are identified. When

filtering small clouds out, the TCC and LWP do not change significantly (less than $-0.05\,\%$ and $-6 \times 10^{-3}\,\mathrm{g\,m^3}$, respectively). A newly condensed cell can be added to the edge of the cloud, linked to a previous cloud cell (Fig. 2, $t_2$), or identified as a new cloud (Fig. 2, $t_3$). The strength of this cloud identification method is that it can identify individual clouds in the model domain quickly.

### 2.2 Description of observational strategy

To improve upon decades of cloud observations, there is a need to follow a cloud throughout its life cycle and determine, with high spatial and temporal resolution, its microphysical and thermodynamical properties. The goal of this study is to derive the best strategy to observe the evolution of an individual cloud. The flight strategy ultimately depends on how long it takes to sample the cloud, which is largely determined by the RPA airspeed to transect the cloud and its turning radius to turn around and re-enter the cloud. In these simulations, the RPA

samples every grid point along its transect. Simulations in this study were conducted using a RPAS airspeed of $15\,\mathrm{m\,s^{-1}}$ and a turning radius of 100 m (Verdu et al., 2019).

In order to optimize the sampling of clouds by the RPA, a Rosette flight pattern is performed in the LES to create a horizontal cross section of the cloud. The flight simulation is controlled by the Paparazzi autopilot module





(Hattenberger et al., 2014), which makes it possible to simulate the behaviour of a RPA within the LES simulation. The autopilot module and the LES are combined with the module CAMS (Cloud Adaptive Mapping System). The payload module which simulates a cloud instrument is embedded in the Paparazzi autopilot module to detect the presence of a cloud using a threshold of LWC $\geq 10^{-2}\,\mathrm{g\,m^{-3}}$. If the LWC threshold is exceeded, the RPA begins its Rosette pattern by conducting a straight line until it exits the cloud. The geometric center (red point in Fig. 3,$t_0$) is calculated after each passage through a cloud. After exiting the cloud (LWC $\leq 10^{-2}\,\mathrm{g\,m^{-3}}$) , the RPA turns back

toward the cloud center (Fig. 3,$t_1$), and the transects are repeated in the form of a Rosette pattern until the cloud disappears.

## 3 Results

This section exploits the high temporal and spatial resolution provided by the LES to optimize the adaptive sampling for static and dynamic cases. First, an overview of the different clouds sampled in the LES is provided before selecting

three clouds representative of the cloud population. Then, an exploration of the selected clouds is carried out with RPA flying off-line in the simulations -first in a static mode (i.e. without taking into account the displacement of the cloud) and then in a dynamic mode (i.e. including the wind advection and time evolution of the cloud).

### 3.1 Overview of a simulated trade cumulus field

During the HFS (12.8km_MNH domain), an average of 300 clouds per output are identified with a minimum of

270 clouds around the 18th minute and a maximum of 350 clouds at the 25th minute (Fig. 4). Individual cloud volumes have been separated into four classes. Class 0 corresponds to a volume between $10^{-4}\,\mathrm{km^3}$ and $10^{-3}\,\mathrm{km^3}$, class 1 between $10^{-3}\,\mathrm{km^3}$ and $10^{-2}\,\mathrm{km^3}$, class 2 between $10^{-2}\,\mathrm{km^3}$ and $10^{-1}\,\mathrm{km^3}$, and class 3 between $10^{-1}\,\mathrm{km^3}$ and $1\,\mathrm{km^3}$ (Table 1). Figure 4 presents the evolution of the number of clouds detected at each time step (every 5 s). The volume distribution shows that one third of the clouds are in class 0, another third in class 1, and another third in

class 2 and 3 with just under 10 clouds exceeding $10^{-1}\,\mathrm{km^3}$ at any given time. The temporal evolution in Fig. 4 of the different classes shows a certain stability in the cloud field.

Some 2150 independent clouds have been identified of which 970 clouds complete a full life cycle within the 30-minute HFS. For clouds with the life cycle fully described, from formation to dissipation, statistics are calculated for thermodynamical and macrophysical properties for each of the four volume classes, as shown in Table 1. For

each class, the minimum and maximum lifetime, cloud base and cloud top are calculated for each cloud over their lifetime and averaged over the total number of clouds for this class.

The cloud base of a newly-formed cloud is always at the level of the LCL, which is around 550 m.asl. The larger the volume, the lower the average cloud base (which ranged between 550-680 m), and inversely, the smaller the volume, the higher the average cloud base (around 800 m). The cloud base also tends to increase when the cloud

dissipates, which increases the average cloud base particularly for small clouds. The lifetime of small clouds is notably





less as they dissipate quickly. The height of the cloud top also increases with the volume, as vertical extension and variations in vertical winds are larger than the horizontal extension for cumulus clouds (Neggers et al., 2003). The larger the volume, the greater the intensity of the downdrafts $w_{min}$ and updrafts $w_{max}$. To calculate $w_{min}$ and $w_{max}$, the highest downdraft and updraft are selected in each individual cloud during its lifetime and then averaged per

class. The maximum of downdrafts and updrafts in this study are observed in the biggest clouds (class 3; $-1.69\,\mathrm{m\,s^{-1}}$ and $2.77\,\mathrm{m\,s^{-1}}$).

The average mass flux of the clouds ($F_m$) is positive, but the standard deviation is larger than the mean in all four classes, highlighting the role of clouds in the transport of water in the atmosphere. The standard deviation ($\sigma$) is 200 times greater than the average flux for cumulus class 0, while it is only 1.37 times greater than the average mass

flux for class 3. Standard deviations of mass flux indicate that variability is related to formation and dissipation. A negative $F_m$ represents the dissipation of the cloud and occurs more often for small clouds than large clouds. The difference in magnitude value of $F_m$ between the size classes is significant, with an order of magnitude of difference in $F_m$ for an order of magnitude change in cloud volume.

### 3.2   Individual cloud description

Inspired by the study of Zhao and Austin (2005a) who use a LES with a similar resolution ($\Delta x = \Delta y = \Delta z = 25$ m) to study thermodynamical processes in individual clouds, with volumes of $10^{-2}\,\mathrm{km^3}$ and $10^{-1}\,\mathrm{km^3}$ , this study focuses on three independent clouds representative of volumes of $10^{-2}\,\mathrm{km^3}$ (N1, class 1), $2 \times 10^{-2}\,\mathrm{km^3}$ to $3 \times 10^{-2}\,\mathrm{km^3}$ (N2, class 2) and $10^{-1}\,\mathrm{km^3}$ (N3, class 3).

#### 3.2.1   Macrophysical and microphysical properties

The microphysical evolution of the life cycle for the three clouds (N1, N2, N3) are followed for 12, 18 and 24 minutes, respectively. The growing phase, corresponding to an increase of volume, comprises 55 % to 65 % of their life cycle. Each of the clouds has a similar cloud-base height (at 550 m), and their cloud top increase follows a logarithmic growth rate, with a higher rate for large clouds. The maximum surface of the horizontal cross section at 150 m above cloud base occurs at t=14 minutes for cloud N1 with $S_{max}= 0.045\,\mathrm{km^2}$, t=10 minutes for cloud N2 with

$S_{max}= 0.28\,\mathrm{km^2}$ and t=15 minutes for cloud N3 with $S_{max} =1.06\,\mathrm{km^2}$. In Fig. 5c, the maximum LWC for the three clouds is compared to their pseudo-adiabatic profile, computed by integrating the adiabatic vertical gradient, $\beta$, through the cloud depth (Korolev, 1993). The difference between pseudo-adiabatic and maximum LWC for each cloud level indicates the degree of entrainment mixing that has occurred. The maximum LWC in the HFS follows the pseudo-adiabatic profile to approximately a third of the height of the clouds. The LWC then remains more or

less constant until decreasing near the summit suggesting higher entrainment rates in the upper part of the clouds.




### 3.2.2 Thermodynamical properties

Consistent with Zhao and Austin (2005b) and Heus et al. (2009), the clouds in the HFS present single or several pulses. As shown in Fig. 5 and Fig. 6, cloud N1 can be described by a simple pulse growth, whereas cloud N2 and N3 show 2 and 3 pulses, the first of which are the most important. The maximum updraft occurs at maximum

volume and at the top of each pulse when the cloud has reached its maturity phase, while maximum downdraft remains relatively constant (Fig. 6a). Similar features are seen for clouds N1, N2 and N3, where the magnitude of the updrafts and downdrafts are related to the size of the cloud (Table 1). Figure 6b represents the time series of the mean vertical mass flux for each cross section. High values of vertical mass flux are located near cloud base and within the cloud core and remain nearly constant up to half the height of the cloud while negative vertical mass

fluxes are always located near the cloud edge and cloud top in the dissipation phase (Fig. 6c).

This individual study of clouds has permitted to describe the heterogeneities of the horizontal and vertical structure of cumulus clouds, in particular with respect to LWC and vertical wind. An observational strategy with sufficiently high sampling resolution is necessary to capture these heterogeneities and is now conducted numerically by embedding the exploration of RPAs in the HFS LES simulation.

### 3.3 Exploration by RPAs in LES

In this subsection, we simulate the capacity of RPAs to explore the horizontal variabilities of the thermodynamic variables in a cloud. First, we demonstrate the concept using cloud N2 in a static state by neglecting its horizontal advection and time evolution. We then repeat for the same cloud N2 but taking into account its evolution with time (dynamic case).

### 3.3.1 Defining a pattern for a static cloud

For this subsection, the cloud is assumed to be static and the flight of the RPA is simulated by embedding the Meso-NH output in the Paparazzi autopilot module. The cloud shape and position as well as thermodynamical variables do not change during the exploration by the RPA. Horizontal wind is fixed to $0\,\mathrm{m\,s^{-1}}$. To demonstrate the viability of the Rosette pattern, described in section 2.2, a cross section at 150 m above N2 cloud base extracted at t=10

minutes is used (corresponding to the time when the cloud reaches its maximum volume)(Fig. 5). The location of the initial entrance in the cloud is random and is shown with a red arrow in Fig. 7a. In this case, the RPA conducts eleven transects at this altitude in the cloud.

After each transect, the sampled horizontal distribution of LWC is reconstructed for the cross section (Fig. 7b). As the Rosette pattern biases sampling of the geometric center, some regions in the cloud N2 are oversampled while

regions toward the cloud edge are not measured at all. Nonetheless, to assess the ability of the Rosette pattern to properly represent the horizontal cross section of the cloud, the probability density functions (PDFs) of reference LWC (Fig. 8a) and w (Fig. 8b) are compared with the PDFs of the reconstructed cloud cross section. The reference





PDF of LWC (black line in Fig. 8a) has a main peak (15 % of cloud grids) at $0.37\,\mathrm{g\,m^{-3}}$, corresponding to the cloud core, and 4 % of grids have a LWC near $0.40\,\mathrm{g\,m^{-3}}$ corresponding to the adiabatic limit at this height above cloud base.

The first transect results in an overestimation of high LWC values, while cloud edges (low LWC values) are underestimated as also shown in (Hoffmann et al., 2014). As the number of transects increases, the LWC biases decrease and the PDF of the LWC approaches the reference distribution.

For vertical winds, the PDF distribution (black line in Fig. 8b) represents a Gaussian distribution with a center located around $0.8\,\mathrm{m\,s^{-1}}$ and representing 15 % of the cloud cross section. The cross section area of downdrafts represents less than 10 % of total vertical wind. High values of updrafts are also overestimated with the first transects; however, the PDF converges to the reference PDF in less time than for LWC. In the study, the practice of only using the RPA observations to map the cloud is called the transect method. To compensate for the above mentioned biases of a single trajectory, simple forms such as a circle or ellipse,provide a simple method to estimate the distribution of LWC and updrafts in the cross section. For example, an equivalent diameter (or the lengths of major and minor axes for an ellipse) can be estimated by an average transect length to retrieve a surface area for a circle or an ellipse. To derive a total LWC ($\mathrm{LWC_{tot}}$) of the cross section, the volume of the reconstructed cloud section is multiplied by the average LWC, $\overline{\mathrm{LWC}}$. The transect method systematically underestimates the cloud volume, while the cloud volumes reconstructed by circle and ellipse methods are more than twice the actual volume of the reference cross section as shown in Fig. 10a. Clearly, none of these relatively simple methods are able to accurately reconstruct the cloud cross section (Fig. 10). To address this deficiency in accurately reconstructing the cloud-cross section using simple methods, we introduce a novel method that uses Gaussian Process Regression (GPR). For following, only transect method (RPA observations) will be compared with Gaussian.

### 3.3.2 Gaussian process regression mapping

Gaussian process regression extends the spatial footprint of an observation by weighting its values with a Gaussian profile.It needs the definition of four length scales ($\lambda_{\mathrm{t,z,y,x}}$), representing spatial (z,y,x) and temporal (t) scales. For the static case, $\lambda_{\mathrm{t}}=\infty$ means that an earlier observation is considered to have the same weight as the last measurements.

To demonstrate the impact of lengthscales on GPR, sensitivity tests are carried out for the cloud N2 using three different lengthscales, corresponding to two, three and four times the resolution of the simulation (50 m, 75 m, 100 m). The differences of the reconstructed map of LWC for the different lengthscales is shown in Fig. 9. For a horizontal lengthscale of 50 m, $\overline{\mathrm{LWC}}$ is underestimated in unsampled regions, consistent with a lengthscale that is too short ($\overline{\mathrm{LWC}}_{\mathrm{reconstructed}} = 0.20\,\mathrm{g\,m^{-3}}$) compared to $\overline{\mathrm{LWC}}_{\mathrm{reference}} = 0.24\,\mathrm{g\,m^{-3}}$). To assess if the cross section of the cloud is correctly defined in its entirety, the root-mean-square error (RMSE) is also calculated and is equal to $8.92\,\mathrm{g\,m^{-3}}$. For a 75 m lengthscale, the reconstructed cross section represents the reference cloud with a $\overline{\mathrm{LWC}}_{\mathrm{reconstructed}} = 0.22\,\mathrm{g\,m^{-3}}$ and a RMSE of $5.71\,\mathrm{g\,m^{-3}}$. For the largest lengthscale, 100 m, the cross section of LWC extends beyond the edges of





the reference cloud, also as expected for a lengthscale that is too large ($\overline{\mathrm{LWC}}_{\mathrm{reconstructed}} = 0.35\,\mathrm{g\,m^{-3}}$).The RMSE for a 100 m lengthscale increases significantly ($14.89\,\mathrm{g\,m^{-3}}$). For the following analysis, the GPR mapping uses a 75 m lengthscale.

### 3.3.3   Criteria for optimizing the exploration

In this section, different methods are applied to better characterize the heterogeneities of the thermodynamic variables and the total LWC. The exploration with Rosette pattern is repeated ten times in the same cloud at the same altitude with different entrances. In each of these explorations, the reconstructed $\mathrm{LWC}_{\mathrm{tot}}$ and $\overline{\mathrm{LWC}}$ is compared to the reference cloud, every 60 seconds for 12 minutes. The reference $\mathrm{LWC}_{\mathrm{tot}}$ and $\overline{\mathrm{LWC}}$ have values of $1.8 \times 10^3\,\mathrm{g}$ and $0.24\,\mathrm{g\,m^{-3}}$, respectively. Three sampling strategies are compared: 1-RPA and 2-RPAs exploration without GPR and 1-RPA exploration with GPR mapping.

The $\mathrm{LWC}_{\mathrm{tot}}$ reconstructed for the three sampling strategies is shown in Fig. 11 with the 1-$\sigma$ dispersion among the ten flights shown as shading. For the first minute of exploration, the three methods underestimate the $\mathrm{LWC}_{\mathrm{tot}}$; however, after the second minute ($\approx$ 2-3 transects), the 1-RPA exploration with GPR method calculates a $\mathrm{LWC}_{\mathrm{tot}}$ close to the reference $\mathrm{LWC}_{\mathrm{tot}}$ while with the two other methods without GPR, the $\mathrm{LWC}_{\mathrm{tot}}$ stays significantly lower than the reference. After the 3rd minute, the GPR method yields a stable $\overline{\mathrm{LWC}}$, within the reference $\overline{\mathrm{LWC}}$ $\pm 10\%$, while 1-RPA and 2-RPA explorations without GPR never attain the reference $\mathrm{LWC}_{\mathrm{tot}}$. In addition, the 1-$\sigma$ variability of $\mathrm{LWC}_{\mathrm{tot}}$ is a factor of three less when using GPR.

Another stated objective of this study is to optimize the sampling strategy in order to best describe the thermodynamical heterogeneities in the cloud. To quantify this, the relative error is calculated as a sum of difference between the reconstructed PDF and the reference PDF of LWC for each of the 20 bins of the distribution as:

$$\mathrm{relative\ \ error} = \frac{1}{nb_{bin}} \sum_{i}^{\mathrm{nb_{bin}}} \frac{|\mathrm{PDF}_{\mathrm{ref,i}} - PDF_{\mathrm{reconstructed,i}}|}{\mathrm{PDF}_{\mathrm{ref,i}}} \tag{1}$$

where $\mathrm{nb_{bin}}$ represents the number of bins, $\mathrm{PDF}_{\mathrm{ref}}$ represents the reference PDF distribution of a variable noted i and $\mathrm{PDF}_{\mathrm{reconstructed}}$ represents the reconstructed PDF distribution with observations for the same variable i.

The relative errors for the three methods are shown in Fig. 12. For the N2 cloud, the time required for the single RPA without GPR to have a relative error below 0.5 is approximatively 350 seconds. With two RPAs without GPR, the time required to have a relative error below 0.5 is reduced to 190 s. Figure 12 shows that one RPA with GPR sufficiently approximates the reference PDF with a relative error below 0.5 in 100 seconds.

The time needed to reach different thresholds relative errors of 10 %, 30 %, 50 % for different variables (LWC, vertical wind and, potential temperature $\theta$) are reported in Table 2 highlighting a significantly improved mapping the cross section by using the GPR method.



As described by Katzwinkel et al. (2014), the growth, maturity and dissipation phases of a cloud life cycle are the
scale of minutes. Consequently, sampling of cloud cross section must also be completed within time scales of a few
minutes. These results show that cloud cross section are sufficiently well represented when using GPR methods.

### 3.3.4 Application for two clouds of different size

In this section, a generalization of the Rosette pattern with the GPR method is applied to static cloud N1 and
N3 at the time they reached their maximum cross sectional area at 150 m above cloud base. Figure 13 shows the
exploration with the Rosette trajectories using GPR mapping. When the dimension of the cloud is smaller than
the turning radius (cloud N1), the exploration is pattern-limited. The relative error remains higher than 0.3 for the
duration of the simulation. When the cloud radius is much larger than the turning radius there is simply more surface
area to sample which prolongs the exploration. For example, the relative error for cloud N3 only approaches 0.2 by
the end of the HFS. Finally, Fig.13 shows that when using GPR, the relative error is below 0.2 midway through the
exploration.

The results demonstrate that the Rosette trajectory associated with a GPR mapping in a static environment is
suitable for sampling thermodynamic and microphysical variables such as LWC or $\theta$, and w. We now assess the
ability to measure thermodynamic and microphysical variables for a case where the cloud evolves with time and
space (i.e., a dynamic case) and reaches 0.1 by the end of the HFS.

### 3.3.5 GPR reconstruction of a cloud evolving with time

In an evolving cloud, the RPA must constantly adjust its trajectory taking into account the advection and spatial
evolution of the cloud. In this subsection, an adaptive exploration follows the cloud in the cloud reference frame. To
demonstrate the challenges in extending the analysis of a static environment (Section 3.3.1) to a dynamic one, the
adaptive exploration is applied here to the cloud N2. For this case, the Rosette pattern (section 2.2) is also applied
at 150 m above cloud base as for the static exploration. The cloud N2 vertical extent reaches 150 m above cloud
base four minutes after its formation, and we start the exploration of the cloud N2 at this time.

The sampling strategy follows the Rosette pattern in the cloud reference frame, where the calculated center of the
cloud moves with respect to the advective wind. The observations of the cloud continues for 12 minutes (Fig. 14),
corresponding to nine transects, until the time of its dissipation (at the level of exploration). The total horizontal
distance traveled is more than 5 km during this period. Figure 14 shows four instances of the dynamic exploration
corresponding to the first, fourth, seventh and eighth transect through cloud N2. The first transect occurs during
the growing phase of the cloud (Fig. 15) and does not cross the region of maximum LWC. As expected, there is a
clear underestimation in the high values of LWC when comparing the PDFs of the LWC (Fig. 15b,1). Between the
fourth and seventh transects (300 to 550 s), the reference cloud is in a relatively stable maturity phase. Once the
cloud center has been established, the exploration is sufficiently efficient to reconstruct a PDF resembling to the
reference case with relative errors between 0.3 and 0.4 (transect 4 and 7 in Fig 15b). However, during the dissipation





phase, the evolution of the cloud's cross section is faster compared to the relatively stable maturity phase and is not well-represented by the PDF (transect 8, Fig 15b), the relative error increases to 0.6. While GPR certainly improve the ability to reconstruction of a cloud's cross section, these results clearly show that to adequately observe the
dissipation phase, the cross section needs to be reconstructed in less time - either via a better sampling strategy of leg adding a second RPA.

## 4   Conclusions

The aim of this study is to determine an observational strategy for reconstructing thermodynamic and microphysical properties of a cross section of a non-precipitating cumulus cloud within a high-resolution Large-Eddy Simulation
(LES). We reproduce a high resolution cumulus cloud field with the Méso-NH model in LES mode (with a 25 meter spatial resolution and a 5 second temporal resolution) derived from the observations in the BOMEX field campaign. The high-resolution simulation (HFS) serves as the basis for this study and compares well to an inter-comparison LES study reported in Siebesma et al. (2003). By applying a novel cloud identification method, we isolated three clouds with different volumes ($10^{-4}\,\mathrm{km^3}$ to $10^{-1}\,\mathrm{km^3}$), representative of the cumulus cloud population. These three
clouds went through a full life-cycle at different times of the HFS and had lifetimes between 12 and 18 minutes. The goal of sampling with a remotely piloted aircraft (RPA) is to reproduce a cloud cross section of liquid water content (LWC) and updraft velocity within a few minutes in order to follow the evolution of a cloud through its growth phase, maturity, and dissipation phases.

An autonomous RPA using a Paparazzi autopilot module is embedded into the LES to conduct an exploration of
a cloud level using a Rosette pattern. In a static environment (a cloud that does not evolve with time), the Rosette pattern is applied to the cloud at the time and the level where its the surface area of its horizontal cross-section is maximum. The Rosette pattern is chosen for the adaptive trajectory since at each exit from the cloud, the RPA automatically conducts a half turn to re-enter the cloud and conducts a subsequent transect through the geometric center of the cloud. The sampling of the cloud continues autonomously using a threshold LWC of $10^{-2}\,\mathrm{g\,m^{-3}}$ to
determine if the RPA has entered or exited the cloud. The geometric center is calculated using a weighted sum of the LWC of from the previous transects. The simulated observations serve to reconstruct different Probability Density Function (PDF) distributions of LWC, vertical winds, volume and total LWC.

Simple methods to derive a cross section using individual observations or assuming a circular or elliptical forms of a cloud do not reproduce key microphysical properties or their variability. Using only the observations from one
or two RPAs underestimates the amount and variability of LWC in the cloud cross-section. Assuming a circle or an ellipse yields a factor of two overestimation of total LWC in the cross-section. We therefore explore another technique to expand the observational footprint using Gaussian Process Regression (GPR). GPR mapping extends individual measurements by applying a confidence level to the surrounding area and time related to a given length scale. The results show that GPR mapping significantly improves the reconstruction of the cloud's cross section. A



sensitivity test of the lengthscale used for GPR mapping indicates that the characteristic scale of 75 m is the best for reconstructing the horizontal LWC in a cloud. In fact, after three transects through the cloud, corresponding to a time of $\approx$ 200 s, the GPR mapping adequately reproduces total LWC (within a relative error of 10 %), as well as the PDF variability of the LWC (within 30 % relative error).

  To extend results of a static exploration to a realistic environment, the Rosette pattern was applied to a cloud
evolving in time and space in a dynamic environment. The GPR mapping method allows to sample the thermodynamical distribution sufficiently well for a cloud during its maturity stage, which is the most stable phase of a cloud life cycle. However, during the growing and dissipating phases, a single RPA coupled with GPR is still insufficient to reproduce the temporal variability of the cloud life-cycle. In order to improve the observational capacity of airborne measurements, various methods are currently being explored, including the use of a camera and increasing the num-
ber of RPAs to reduce the time for reconstructing a cloud's cross section. To optimize the dynamic exploration of a cloud, exploration patterns with a different trajectories RPA, flight characteristics (such as airspeed and turning radius), as well as coordination between multiple RPAs constitute necessary steps in improving our ability to observe the cloud life cycle.

*Data availability.* Data for the static case are currently being archived and will be accessible online. Due to size of the data
for the dynamic simulation (1.5 TB), please contact the corresponding authors.

*Author contributions.* N.M conducted the analysis of the data and wrote the paper. G.C supervised the project, verified the analytical methods and edited the paper. F.C carried out the simulation, verified the analytical methods and edited the paper. N.V co-developed cloud identification method. T.V, G.H have designed RPA patterns for the Paparazzi autopilot. P.N, S.L conceived and developed the Cloud Adaptive Mapping System (CAMS) module.

*Competing interests.* The authors declare that they have no conflict of interest.

*Acknowledgements.* This research was supported by Agence Nationale de la Recherche (Project-ANR-17-CE01-0003), Aerospace Valley and Météo-France. The simulations were performed using the supercomputer Beaufix of Météo-France in Toulouse, France. Cloud identification method co-developed by N.V is available on $https : //gitlab.com/tropics/objects/$.

  The CAMS module is available on $https : //redmine.laas.fr/projects/nephelae-devel/wiki$ .The Paparazzi module on
$https : //wiki.Paparazziuav.org/wiki/Main\_Page$.





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





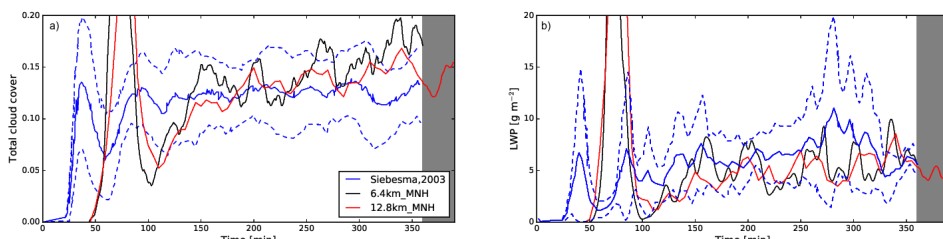

**Figure 1.** Temporal evolution of a) total cloud cover and b) liquid water path from the Siebesma et al. (2003) intercomparison (blue lines, solid line for the mean and dotted line for the $\overline{1-\sigma}$ standard deviation), black line for 6.4km_MNH domain and red line for 12.8km_MNH domain. Grey shaded area corresponds to Meso-NH high frequency outputs (HFS).

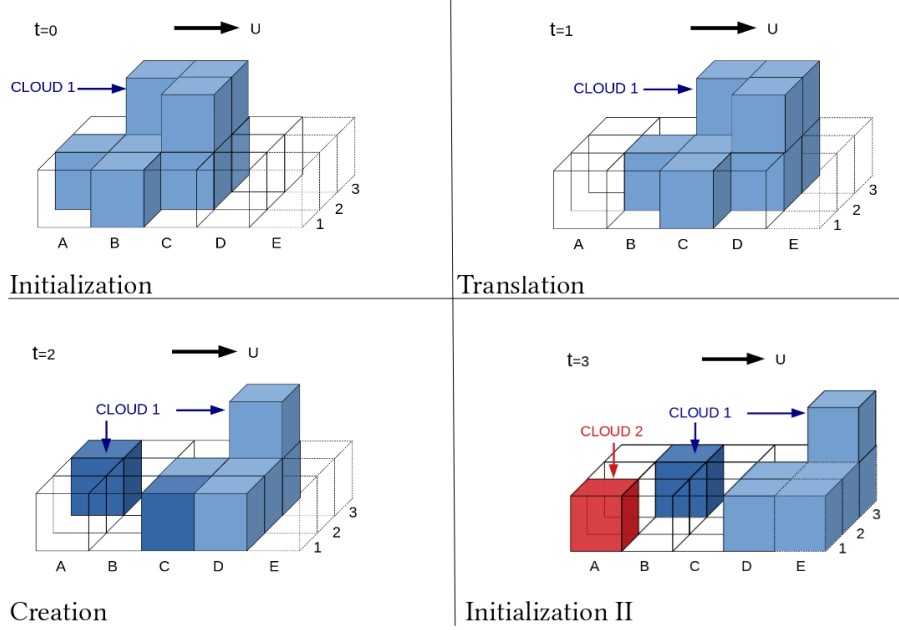

**Figure 2.** Scheme of temporal cloud identification method for following a cloud in a dynamic environment

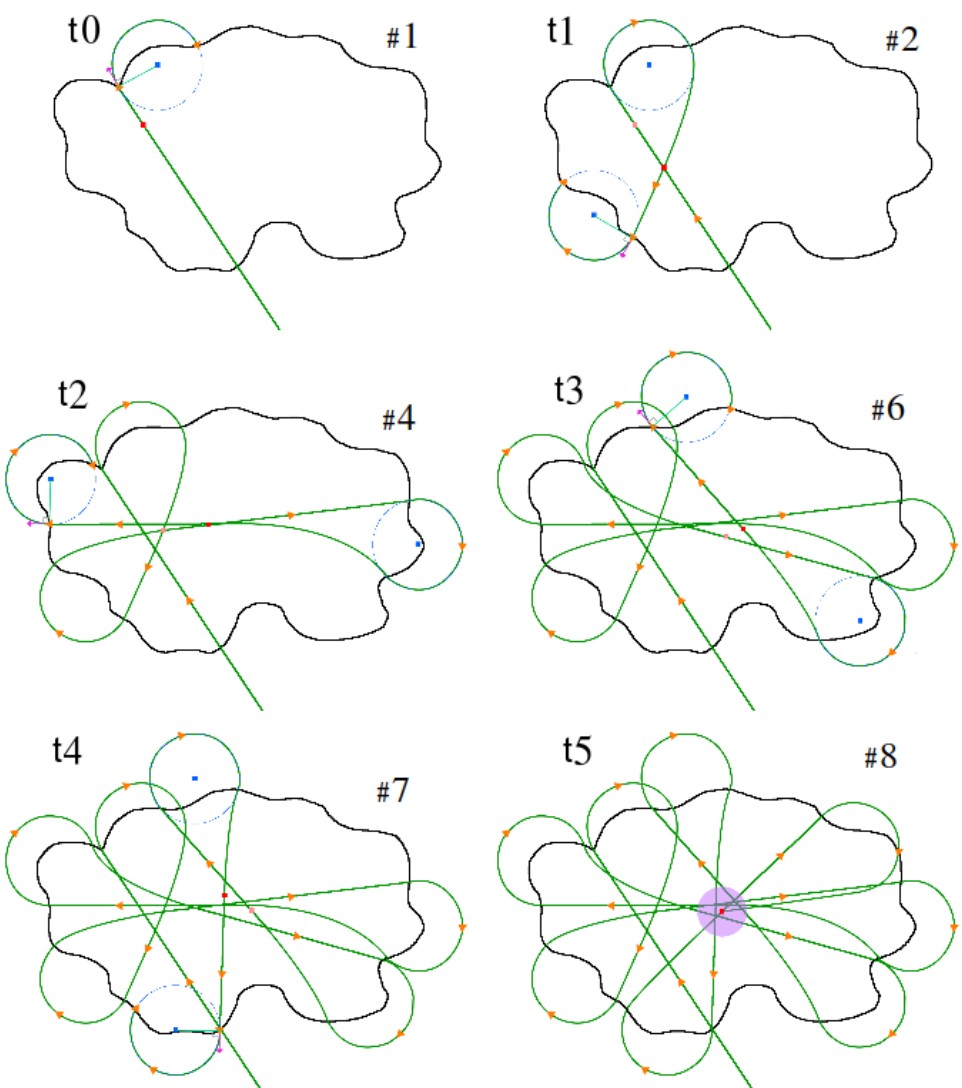

**Figure 3.** Rosette pattern for different times (and the number of transect associated) of exploration, adapted from Verdu et al. (2019). Green lines represent the RPA trajectories, red points the calculated geometric center at different times of exploration, purple point the last geometric center.





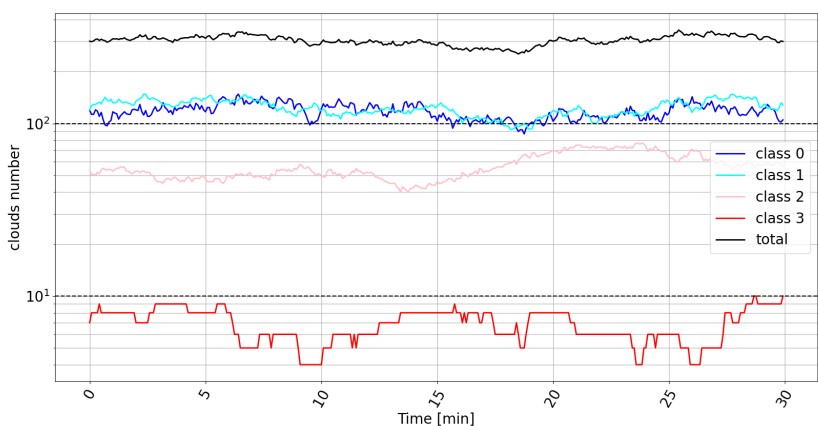

**Figure 4.** Temporal evolution of the number of clouds at each model output classified by their different volumes from Table 1.

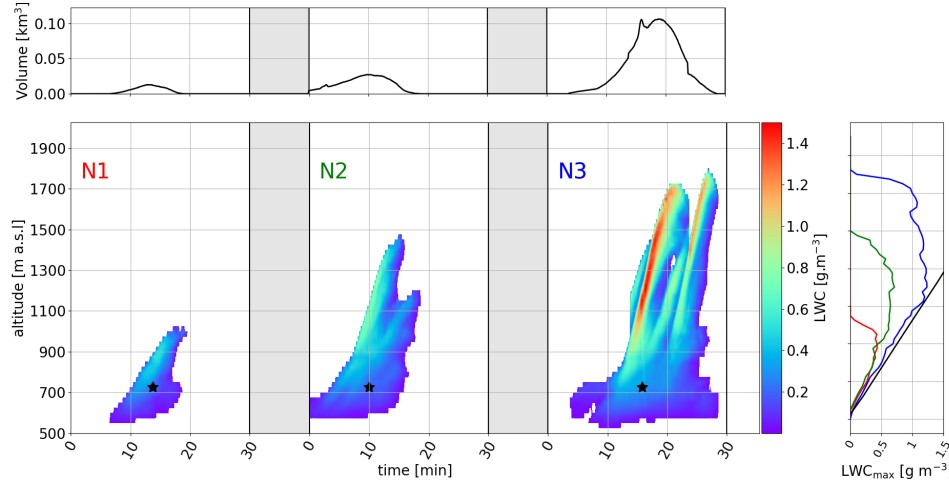

**Figure 5.** Life cycle of three clouds N1,N2,N3 for a) their volume and b) their LWC. The black star represents altitude and time for static cloud exploration. c) shows the comparison between the maximum LWC for each vertical level during each life cloud cycle and the limit of pseudo-adiabatic LWC (black line).



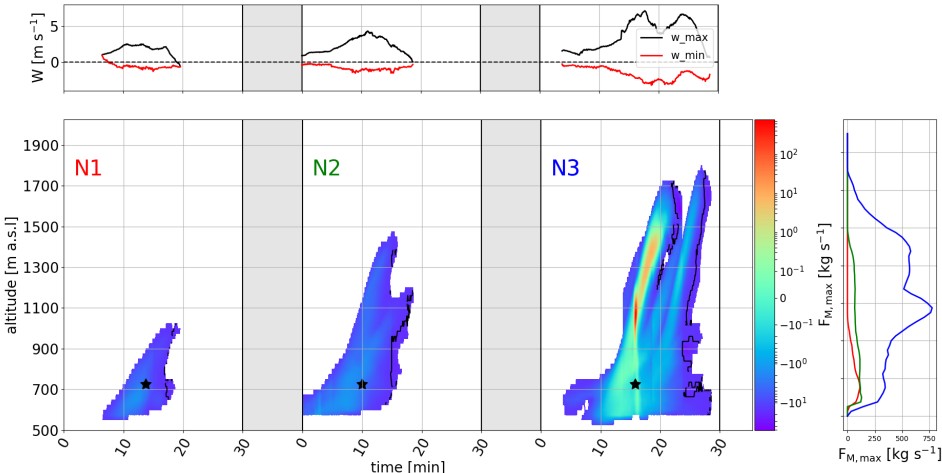

**Figure 6.** Life cycle of three clouds N1,N2,N3 for a) their minimum and maximum vertical wind, b) their mass flux and c) vertical profile of mass flux for each cloud. The black line in b) corresponds to negative vertical flux. The black star represents altitude and time for static cloud exploration.

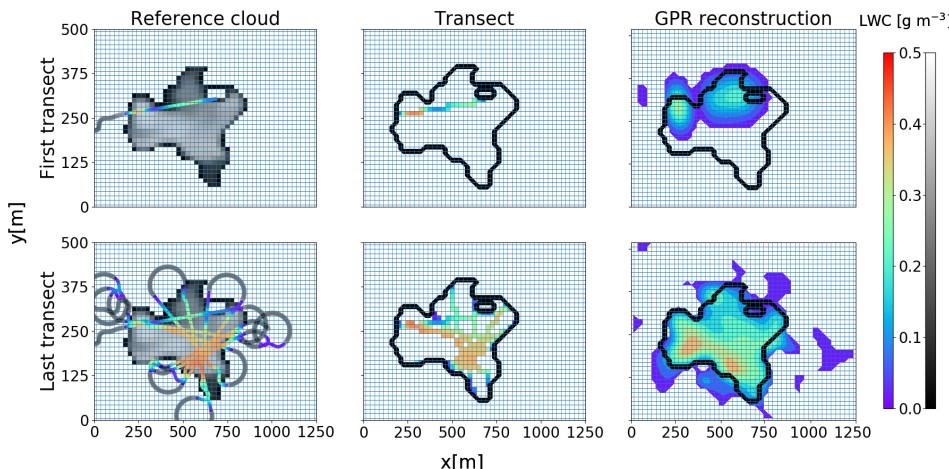

**Figure 7.** a) Cross section of simulated cloud N2 at 150 m above cloud base in gray with transect of the RPA where the color represents the LWC measured. b) Reconstructed LWC in the cross section based on RPA transects. c) Reconstructed LWC in the cross section with GPR mapping with $\lambda_x$=75 m. The first row corresponds to the end of the first transect, the second row corresponds to all transects at the end of the exploration. The red arrow represents the first entrance in cloud N2 starting the Rosette pattern (section 2.2)





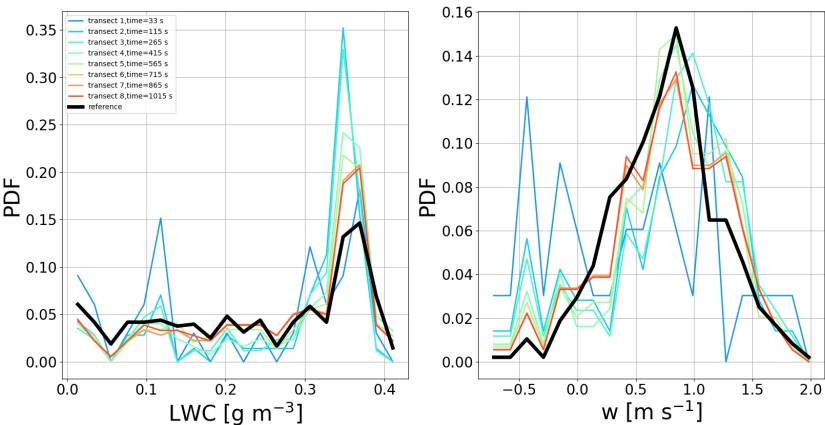

**Figure 8.** Reconstructed probability density function (PDF) of a) LWC with time (color) compared to the reference of cloud N2 (black) at alt=700 m. b) Same for vertical wind.

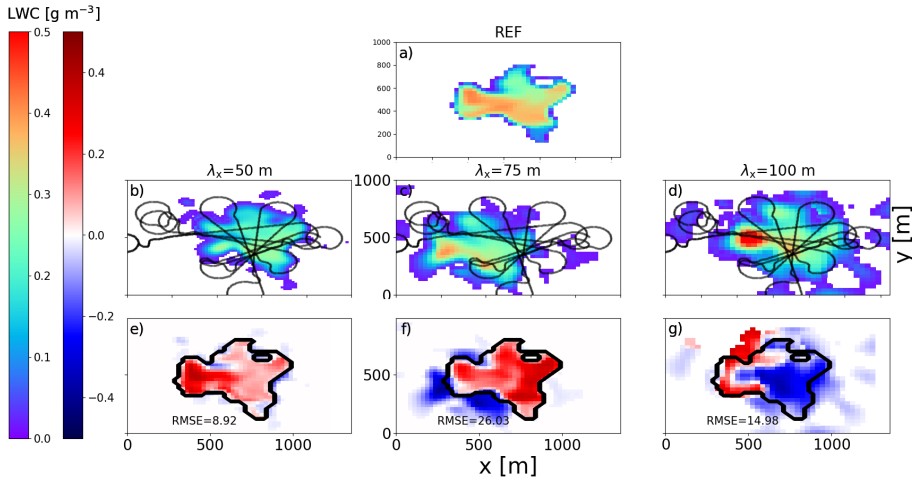

**Figure 9.** a) The first row corresponds to a cross section of simulated cloud N2 at 150 m above the cloud base in color shade. The second row corresponds to a cross section of LWC reconstructed for the cloud N2 with GPR for b) $\lambda_x = 50$ m, c) $\lambda_x = 75$ m, d) $\lambda_x = 100$ m. The third row corresponds to the difference between reference cross section and reconstructed cross section by GPR for the three lengthscales, RMSE is also shown.





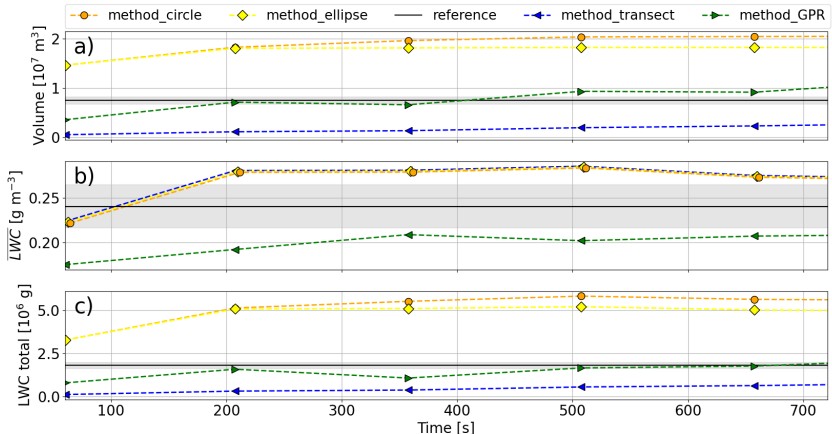

**Figure 10.** a) Reconstructed cloud volume with circle method (blue line), ellipse method (green line), transect method (red line) and GPR method (orange line) compared to the reference volume (black line) for cloud N2 in static state at 700 m for one exploration b) Blue line corresponds to $\overline{LWC}$ calculated by transect method, green line by GPR method and dotted black line the reference $\overline{LWC}$ c) Same for integration of LWC in the section volume.

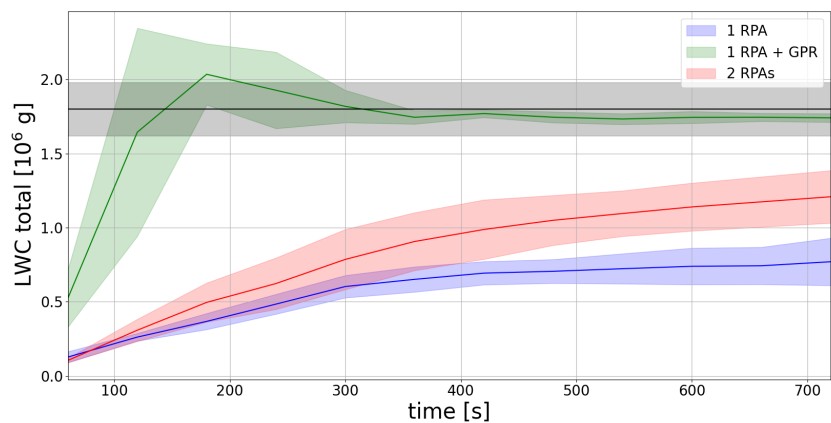

**Figure 11.** Temporal evolution of reconstructed LWC total with the transect method with one RPA (blue line) or two RPAs (red line) and with the GPR method with $\lambda_x$=75 m (green line) for a single RPA. Colored shading areas represent LWC$_{tot}$ $\pm$ 1-$\sigma$ for the ten explorations for each 60 s. The black line corresponds to reference LWC$_{tot}$ in the cloud N2 section in static state at 700 m and shaded grey area corresponds to $\pm$ 10 % of LWC$_{tot}$.





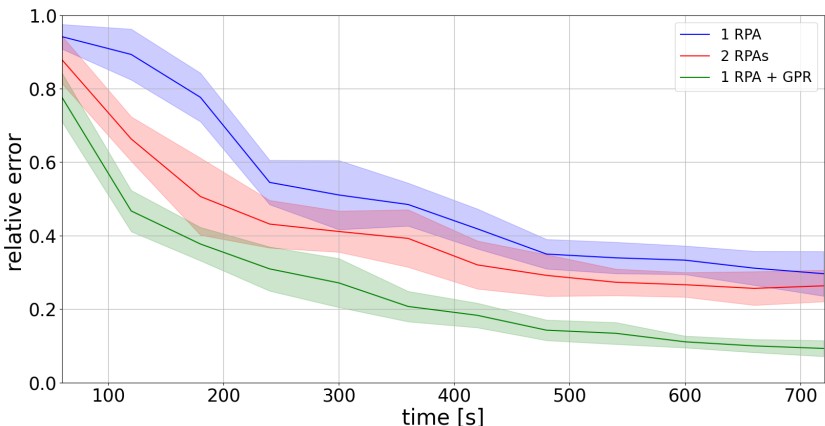

**Figure 12.** Temporal evolution of relative error in PDF of LWC distribution for single RPA with transect method (blue line), two RPAs with transect method (red line) and GPR method (green line) during the exploration of cloud N2 in static state.

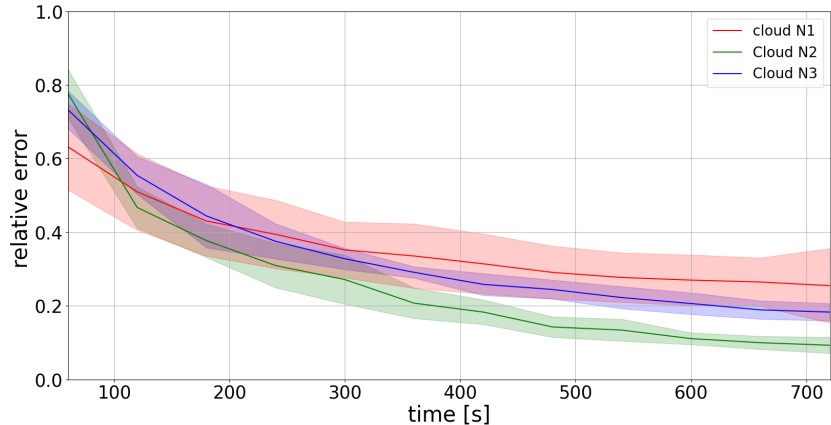

**Figure 13.** Temporal evolution of relative error of LWC distribution for cloud N1 (red line),cloud N2 (green line) and cloud N3 (blue line) in static state. Shaded area corresponds to $\overline{1-\sigma}$ for 5 explorations.


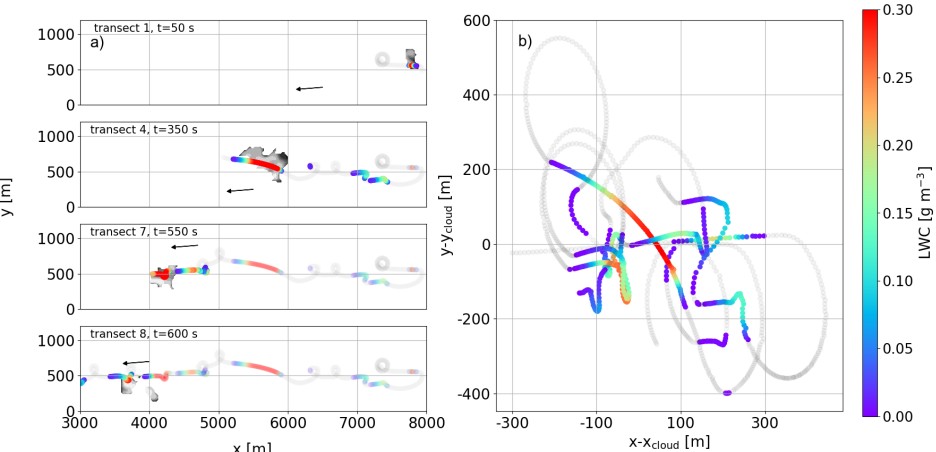

**Figure 14.** a)Trajectories of exploration by following the cloud N2 for four different transects. The colors represent the measured LWC and grey surface corresponds to cross section of reference cloud N2 for the different times. The shading colors correspond to the past transects. The black arrow represents the direction of the advective wind. b) Measured LWC in cloud frame during the exploration. The colored lines correspond to the values of measured LWC exceeding $10^{-2}\,\mathrm{g\,m^{-3}}$ and the grey lines, the measured LWC below this threshold.

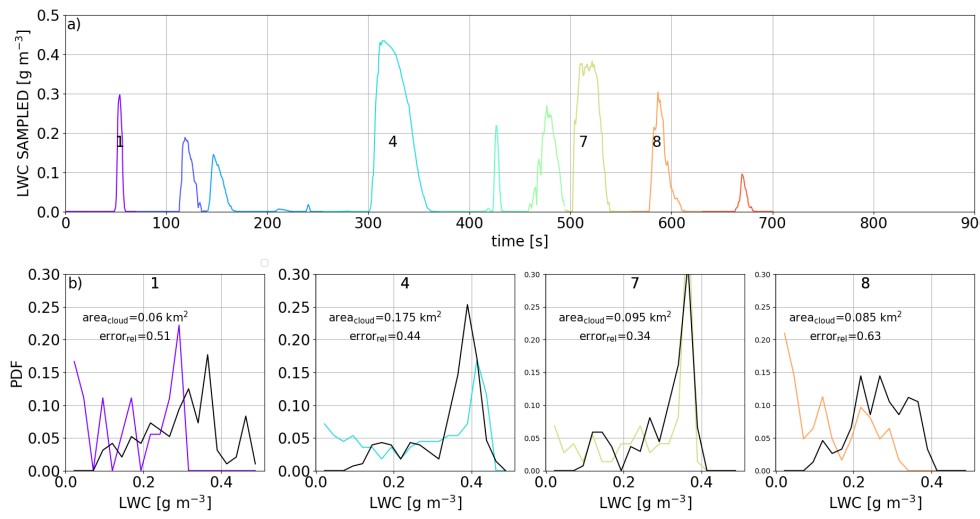

**Figure 15.** a) Temporal evolution of LWC measured by RPA for different transects (colors). b) Reconstructed probability function of LWC in color and by the reference in black for four transects of exploration (transect 1,4,7,8).


**Table 1.** Min, max, mean and ($1$-$\sigma$) standard deviation of macroscopic and dynamic characteristics of the 970 clouds that complete a full life cycle during HFS.

| class | volume $[\mathrm{km}^{-3}]$ | lifetime [min] min/max | cloud$_{base}$ [m] min/max | cloud$_{top}$ [m] min/max | $w_{min}$ [m s$^{-1}$] mean $\pm$ ($\sigma$) | $w_{max}$ [m s$^{-1}$] mean $\pm$ ($\sigma$) | $F_m$ [kg s$^{-1}$ m$^{-2}$] mean $\pm$ ($\sigma$) | n |
|---|---|---|---|---|---|---|---|---|
| 0 | $10^{-4}$ -$10^{-3}$ | 1.16 /9.16 | 812/850 | 848/877 | -0.3 $\pm$ 0.35 | 0.32 $\pm$ 0.41 | 0.03 $\pm$ 6.01 | 501 |
| 1 | $10^{-3}$ -$10^{-2}$ | 3.55/ 17.75 | 816/943 | 935/1035 | -0.61 $\pm$ 0.35 | 0.66 $\pm$ 0.45 | 1.65 $\pm$ 60.03 | 333 |
| 2 | $10^{-2}$ -$10^{-1}$ | 5.93 /17.08 | 686/872 | 1015/1327 | -1.01 $\pm$ 0.43 | 1.25 $\pm$ 0.52 | 160 $\pm$ 440 | 118 |
| 3 | $10^{-1}$ -1 | 7.44 /21.58 | 555/680 | 1518/1634 | -1.69 $\pm$ 0.41 | 2.77 $\pm$ 0.84 | 1245 $\pm$ 1707 | 18 |

**Table 2.** Time required to calculate relative error inferior to 10 %,30 % and 50 % for the LWC, potential temperature, and vertical wind,W, of reference cloud and LWC$_{tot}$ for 1-RPA, 2-RPA and 1-RPA+GPR exploration.

| | relative error (time [s]) | | | | | | | | | LWC tot (time [s]) | | |
|---|---|---|---|---|---|---|---|---|---|---|---|---|
| | LWC | | | $\theta$ | | | w | | | | | |
| | 1-RPA | 2-RPA | GPR | 1-RPA | 2-RPA | GPR | 1-RPA | 2-RPA | GPR | 1-RPA | 2-RPA | GPR |
| 10 % | - | - | 658 | - | - | 574 | - | - | - | - | - | 210 |
| 30 % | 726 | 463 | 255 | 480 | 223 | 186 | 715 | 561 | 324 | - | - | 99 |
| 50 % | 325 | 185 | 113 | 229 | 132 | 74 | 420 | 265 | 131 | - | 356 | 80 |