# Peer review of "Use of Large-Eddy simulations to design an adaptive sampling strategy to assess cumulus cloud heterogeneities by Remotely Piloted Aircraft"

_Atmospheric Measurement Techniques, 2021_

## Referee Comment (RC1)

**Review of "Use of Large-Eddy simulations to design an adaptive sampling strategy to assess cumulus cloud heterogeneities by Remotely Piloted Aircrafts" by Maury et al. (amt-2021-20)**

The manuscript assesses how Remotely Piloted Aircrafts (RPAs) could be applied to measure the spatial and temporal distribution of liquid water in shallow cumulus clouds. Within high-resolution large-eddy simulations (LESs), virtual RPAs mimic measurement patterns, which allow the authors to evaluate the potential of this promising measurement approach for static and a temporally developing cloud.

Overall, the study is very interesting, and the presented approach could be beneficial to measure small-scale heterogeneities in clouds. However, the manuscript requires substantial major revisions to present a convincing sampling strategy and to reach a state acceptable for publication in Atmospheric Measurement Techniques. More details follow below.

**Major Revisions**

*Effective resolution.* The authors need to determine an effective resolution of the suggested sampling strategy to assess whether it can measure small-scale heterogeneities in clouds. The mixing timescale $\tau_{mix} = (l^2/\varepsilon)^{1/3}$ might be useful here (Baker et al. 1984). $\tau_{mix}$ indicates how fast a heterogeneity with a typical lengthscale $l$ decays to the Kolmogorov lengthscale for a turbulence kinetic energy dissipation rate ε. Figure 11 shows that for estimating the total LWC, at least 300 s of measurements are necessary (1 RPA + GPR). If you equate this to the mixing timescale and solve for $l$, one yields an effective resolution of 164 m. (I used $\varepsilon = 10 \text{ cm}^2 \text{ s}^{-3}$, which is a low but typical value for shallow cumulus; more accurate estimates of ε might be available from the LES model's subgrid scheme.) Accordingly, only heterogeneities larger than 164 m can be assessed reasonably, which is probably too large to investigate the dynamics of entrainment and mixing, which are associated with lengthscales ≤ 100 m (e.g., Bodenschatz et al. 2010). This quick calculation indicates that several RPAs, which can shorten the time to measure the cloud, are necessary to gain reasonable insights into the small-scale dynamics of clouds, and the authors have missed the opportunity to include more RPAs in their analysis.

*The use of multiple RPAs.* The number of RPAs is probably insufficient to resolve the cloud at spatial scales relevant to entrainment and mixing, as indicated above. Furthermore, the authors also conclude that one RPA is insufficient for assessing a developing cloud (ll. 348 – 351). Accordingly, why do the authors not investigate the impact of a potentially much larger number of RPAs? The presented workflow for determining the virtual RPA measurements could be repeated several times with different initial locations without too much additional work. The results would probably assess how the results improve as a function of the number of RPAs, and how many RPAs are at least required to sample a developing cloud. This information is highly relevant for planning real applications of RPA sampling, and need to be included in this manuscript.

**Minor Revisions**

L. 5: Why are microscopic properties disregarded here?

Ll. 30 – 31: Clarify "climate responses".

L. 32: Entrainment and mixing create heterogeneities.

Ll. 36 – 39: Sub-grid scale liquid water content is not a quantity that is predicted in most LES models.

L. 42: A more appropriate reference for CARRIBA is Siebert et al. (2013).

L. 47: Regarding the oversampling of the cloud core, you should cite Hoffmann et al. (2014) here.

L. 54: As above, sub-grid scale variations in the cloud droplet number are rarely predicted in LES.

L. 98: Why did you use only a one-moment microphysics scheme? Turbulent mixing, which is a potential subject to be addressed within the presented framework (l. 32), is known to change the

droplet number significantly (e.g., Baker and Latham 1979), which requires a two-moment microphysics scheme to be represented correctly.

L. 100: In the literature, the term "saturation adjustment" is more frequently used than "all-or-nothing". Please consider changing.

Ll. 104 – 107: What do you expect from the larger domain? Why do you introduce that smaller domain at all?

L. 109: The term "High Frequency Simulation" is misleading. The simulation is not high frequent; the output of data is. I suggest "High Frequency Sampling" as an alternative.

Ll. 116 – 117: It is well known that the TCC increases for higher resolutions. I suggest citing Matheou et al. (2011) here.

L. 158: How do you calculate the geometric center? The red dot in Fig. 3 does not look very much in the geometric center. In the conclusions, you state that the geometric center is weighted by the LWC, which is relevant information but should be stated here already.

Ll. 192 – 193: Clarify how the large standard deviation highlights the role of clouds in the transport of water in the atmosphere.

L. 204: How do you define microphysical properties?

Ll. 258 – 268, 373 – 376: Is the discussion of simple geometric forms necessary? I would omit these lines in the revised manuscript. Why do you address Fig. 10 before Fig. 9?

Fig. 10: The colors stated in the caption do not correspond to the colors assigned in the line labels.

L. 290: Clarify what do you mean by 1-RPA and 2-RPA. I assume that the latter describes the investigation with two RPAs, but it is not stated explicitly.

Fig. 11: It might be helpful to add the number of transects on the x-axis, in addition to time.

Ll. 324 – 325: Does this statement refer to N2? This is, however, already shown in Fig. 12. Please clarify.

L. 333: The static cloud has been discussed in the Sections 3.3.1 to 3.3.4.

L. 340: For clarity, consider calling the four "instances" "timeframes".

Fig. 14b: Why is this panel not discussed in the text? And why are there two sets of starting and ending points?

Ll. 350 – 351: A potentially better sampling strategy has not been discussed. Omit this sentence.

**Technical Corrections**

Ll. 2 and 26: Decide on "earth" or "Earth".

L. 14: "maritime" instead of "oceanic".

L. 19: "distribution", not "distributions".

Ll. 133 – 142: The figure uses a slightly different notation for the points in time. (E.g., t=0 and not t0.)

L. 150: "RPA" not "RPAS".

Ll. 221 – 225: Where are the panels a to c in Fig. 6?

L. 241: Where is the red arrow in Fig. 7a? And where is Fig. 7a?

L. 252: Check the citation style: "Hoffmann et al. (2014)" instead of "(Hoffmann et al. 2014)".

L. 271: Add a blank after "profile."

L. 278: There is one parenthesis ")" too many.

**References**

Baker, M. B., & Latham, J. (1979). The evolution of droplet spectra and the rate of production of embryonic raindrops in small cumulus clouds. *Journal of the Atmospheric Sciences*, *36*(8), 1612-1615.

Baker, M. B., Breidenthal, R. E., Choularton, T. W., & Latham, J. (1984). The effects of turbulent mixing in clouds. *Journal of Atmospheric Sciences*, *41*(2), 299-304.

Bodenschatz, E., Malinowski, S. P., Shaw, R. A., & Stratmann, F. (2010). Can we understand clouds without turbulence?. *Science*, *327*(5968), 970-971.

Hoffmann, F., Siebert, H., Schumacher, J., Riechelmann, T., Katzwinkel, J., Kumar, B., Götzfried, P. & Raasch, S. (2014). Entrainment and mixing at the interface of shallow cumulus clouds: Results from a combination of observations and simulations. *Meteorologische Zeitschrift 23 (2014)*, *23*(4), 349-368.

Matheou, G., Chung, D., Nuijens, L., Stevens, B., & Teixeira, J. (2011). On the fidelity of large-eddy simulation of shallow precipitating cumulus convection. *Monthly Weather Review*, *139*(9), 2918-2939.

Siebert, H., Beals, M., Bethke, J., Bierwirth, E., Conrath, T., Dieckmann, K., ... & Wex, H. (2013). The fine-scale structure of the trade wind cumuli over Barbados–an introduction to the CARRIBA project. *Atmospheric Chemistry and Physics*, *13*(19), 10061-10077.

---

## Author Comment (AC1)

Review of "Use of Large-Eddy simulations to design an adaptive sampling strategy to assess cumulus cloud heterogeneities by Remotely Piloted Aircrafts" by Maury et al. (amt-2021-20)

The manuscript assesses how Remotely Piloted Aircrafts (RPAs) could be applied to measure the spatial and temporal distribution of liquid water in shallow cumulus clouds. Within high-resolution large-eddy simulations (LESs), virtual RPAs mimic measurement patterns, which allow the authors to evaluate the potential of this promising measurement approach for static and a temporally developing cloud. Overall, the study is very interesting, and the presented approach could be beneficial to measure small-scale heterogeneities in clouds. However, the manuscript requires substantial major revisions to present a convincing sampling strategy and to reach a state acceptable for publication in Atmospheric Measurement Techniques. More details follow below.

We wish to thank the reviewer for his/her careful review. Below are our responses (in red) to the comments (in black) on a point-by-point basis. The text that has changed in the manuscript is indicated in quotation marks.

**Major Revisions**

Effective resolution. The authors need to determine an effective resolution of the suggested sampling strategy to assess whether it can measure small-scale heterogeneities in clouds. The mixing timescale $\tau_{mix}$= $(l^2/\epsilon)^{(1/3)}$' might be useful here (Baker et al. 1984). $\tau_{mix}$ indicates how fast a heterogeneity with a typical lengthscale $l$ decays to the Kolmogorov lengthscale for a turbulence kinetic energy dissipation rate $\epsilon$. Figure 11 shows that for estimating the total LWC, at least 300 s of measurements are necessary (1 RPA + GPR). If you equate this to the mixing timescale and solve for $l$, one yields an effective resolution of 164 m. (I used $\epsilon$ = 10 cm² s-3, which is a low but typical value for shallow cumulus; more accurate estimates of $\epsilon$ might be available from the LES model's subgrid scheme.) Accordingly, only heterogeneities larger than 164 m can be assessed reasonably, which is probably too large to investigate the dynamics of entrainment and mixing, which are associated with lengthscales $\leq$ 100 m (e.g., Bodenschatz et al. 2010). This quick calculation indicates that several RPAs, which can shorten the time to measure the cloud, are necessary to gain reasonable insights into the small-scale dynamics of clouds, and the authors have missed the opportunity to include more RPAs in their analysis.

We appreciate the reviewer's insight and fully agree that a scale analysis establishes criteria for a sampling strategy. The mixing time stated Baker et al.1984, provides a target temporal resolution for assessing the evolution within the cloud. As noted by the reviewers, for the simulations shown in Figure 11, this scale analysis suggests that total LWC needs to be measured within temporal scales of ca. 200 sec (using a 100 m length scale defined in Bodenschatz et al., 2010).

We can also assess mixing length scales (L) using the follow expression (Taylor, 1935),

$L \sim w3 / \varepsilon$

Updrafts ($w$) in trade wind cumuli are typically > 1 m/s (Katzwinkel et al., 2014), resulting in length scales > 1000 m (which are often larger than the cloud itself). This scale analysis suggests that spatial scales in clouds are driven by gradients in updraft, which mostly occur at the cloud edges. Therefore, to estimate total LWC, the identification of cloud edge and fractal morphology of the cloud is most important. This is consistent with the analysis presented in Figure 9 that reconstructing clouds using circles or ellipses are inadequate -- and that using GPR to map horizontal cross sections of a cloud to estimate total LWC is a viable approach.

[Figure]

Improvements in the sampling strategy can be accomplished in multiple ways, including by adding RPAs, improving the trajectory paths associated with the autonomous sampling, and optimizing the GPR length scales. We have since added the results from the simulation of two RPAs (Fig. 11) and show that the time to estimate the total LWC approaches 200 sec -- a temporal resolution that is sufficient for following the evolution of a cloud. We expect optimization of GPR length scales and using coordinated trajectories will improve the mapping of the cloud cross section and reduce the time to estimate the total LWC.

The use of multiple RPAs. The number of RPAs is probably insufficient to resolve the cloud at spatial scales relevant to entrainment and mixing, as indicated above. Furthermore, the authors also conclude that one RPA is insufficient for assessing a developing cloud (ll. 348 – 351). Accordingly, why do the authors not investigate the impact of a potentially much larger number of RPAs? The presented workflow for determining the virtual RPA measurements could be repeated several times with different initial locations without too much additional work. The results would probably assess how the results improve as a function of the number of RPAs, and how many RPAs are at least required to sample a developing cloud. This information is highly relevant for planning real applications of RPA sampling, and need to be included in this manuscript.

The GPR mapping with the use of two RPAs was carried out, and has been added to the new version of the manuscript (L301-306; purple line in Fig.11 and Fig.12). As mentioned in the previous response, adding two RPAs with GPR reduced the time to estimate total LWC from 300 seconds for 1 RPA+GPR to ca. 200 seconds.

We intend to investigate GPR mapping using a much larger number of RPAs; however, exponentially more computer resources were needed and, unfortunately, we were not able to go beyond two RPAs for this study. We are currently optimizing the simulations to conduct the cloud mapping with several (up to 6) RPAs to define temporal scales as a function of the number of RPAs.

Minor Revisions

L. 5: Why are microscopic properties disregarded here?

We include a reference to microphysical properties. The text now reads "… thermodynamic, microphysical, and macroscopic … "

Ll. 30 – 31: Clarify "climate responses".

We have contributed the role of cumulus clouds to the radiative budget on Earth.

The text now reads "... the impacts of such clouds on the climate radiation budget."

L. 32: Entrainment and mixing create heterogeneities.

We have specified the two types of heterogeneities to the text. The text now read "Mixing process and entrainment impact cloud microphysical properties…"

Ll. 36 – 39: Sub-grid scale liquid water content is not a quantity that is predicted in most LES models. L.54: As above, sub-grid scale variations in the cloud droplet number are rarely predicted in LES.

We agree and we had already acknowledged in the text that LES do not reproduce sub-grid heterogeneities.

"However such models, with a horizontal resolution of a few tens of meters, still use parameterizations to represent cloud microphysics and small-scale turbulence to correctly reproduce sub-grid heterogeneities inside cumulus clouds such as sub-grid scale liquid water content (LWC) variability resulting from mixing processes at the cloud-air interface."

L. 42: A more appropriate reference for CARRIBA is Siebert et al. (2013).

Corrected

L. 47: Regarding the oversampling of the cloud core, you should cite Hoffmann et al. (2014) here. Corrected

L. 98: Why did you use only a one-moment microphysics scheme? Turbulent mixing, which is a potential subject to be addressed within the presented framework (l. 32), is known to droplet number significantly

(e.g., Baker and Latham 1979), which requires a two-moment microphysics scheme to be represented correctly.

We agree, but the reason for using a one-moment scheme is that no aerosol size distributions were measured during the BOMEX campaign to initialize a two-moment microphysical scheme.

In addition, since these RPAs did not measure aerosol size distribution, we chose to stay with the use of a one-moment scheme to compare with previous BOMEX simulations (Siebesma et al., 2003).

L. 100: In the literature, the term "saturation adjustment" is more frequently used than "all-or- nothing". Please consider changing.

Corrected

Ll. 104 – 107: What do you expect from the larger domain? Why do you introduce that smaller domain at all?

The Siebesma et al., 2003 intercomparison exercise was based on a 6.4 km square domain. We first used this domain to validate our model and then we enlarged the domain by (2 x 2 -- a factor of four) to have a larger number of clouds and a more representative cloud population for subsequent analysis. The larger domain facilitates exploration by the RPAs without constraints from domain boundaries.

L. 109: The term "High Frequency Simulation" is misleading. The simulation is not high frequent; the output of data is. I suggest "High Frequency Sampling" as an alternative.

Corrected

Ll. 116 – 117: It is well known that the TCC increases for higher resolutions. I suggest citing Matheou et al. (2011) here.

We thank the reviewers for pointing this out. We have added the citation.

L. 158: How do you calculate the geometric center? The red dot in Fig. 3 does not look very much in the geometric center. In the conclusions, you state that the geometric center is weighted by the LWC, which is relevant information but should be stated here already.

Yes, the geometric center is weighted by the LWC from the cloud mapping. We have included this in the manuscript "using a weighted sum of the LWC".

Indeed, during the first transects, the geometric center is still incorrectly placed but at the end of the exploration, the geometric center is located in the correct place. This has been clarified in the text as well.

Ll. 192 – 193: Clarify how the large standard deviation highlights the role of clouds in the transport of water in the atmosphere.

We had not intended to relate the large standard deviation to the role of clouds in the transport of water in the atmosphere, but to the positive mass flux. This phrase is not needed and has been removed.

L. 204: How do you define microphysical properties?

We have removed the term "microphysical", because it is not appropriate here.

Ll. 258 – 268, 373 – 376: Is the discussion of simple geometric forms necessary? I would omit these lines in the revised manuscript. Why do you address Fig. 10 before Fig. 9?

We feel that the discussion of simple geometric forms is necessary. Certain studies as Rodts et al., 2003 made the direct observation of cumulus, and proposed the circular or ellipsoidal shape of the cumulus sections. Figure 10 (previously Figure 9) clearly shows that mapping by GPR improves the estimate of a total LWC.

Also, we have changed the order of the figures.

Fig. 10: The colors stated in the caption do not correspond to the colors assigned in the line labels.

Corrected

L. 290: Clarify what do you mean by 1-RPA and 2-RPA. I assume that the latter describes the

investigation with two RPAs, but it is not stated explicitly.

We clarified this point by adding the following text to the manuscript: "Four sampling strategies are compared: single RPA exploration just using observations along the trajectory (1-RPA) and with GPR mapping (1-RPA + GPR). Similar notation is used for the 2-RPA exploration."

Fig. 11: It might be helpful to add the number of transects on the x-axis, in addition to time.

The transects for each of the different explorations do not have the same duration, which led us to express the x-axis in time rather than the number of transects.

Ll. 324 – 325: Does this statement refer to N2? This is, however, already shown in Fig. 12. Please

clarify.

Corrected

The manuscript now reads: " Finally, Fig.13 shows that when using GPR for a middle cloud (cloud N2), the relative error is below 0.2 midway through the exploration."

L. 333: The static cloud has been discussed in the Sections 3.3.1 to 3.3.4.

We have corrected the section numbers.

L. 340: For clarity, consider calling the four "instances" "timeframes".

We agree and have changed the text in the manuscript.

Fig. 14b: Why is this panel not discussed in the text? And why are there two sets of starting and

ending points?

We have added the description for Fig 14b.

"Figure 14b represents the RPA transects in a fixed frame where advection has been removed (in a Langrangian reference frame).  The cloud transects are 500-600 m long, and map the evolution of the cloud's boundary."

The flights of RPAs took place at a simulated altitude of 700 m above the ground. If the altitude of the drone exceeded an altitude range of +/- 10 m, the simulated measurements were from an overlying or underlying mesh and not recorded. This means that in some places, the points are not plotted, especially during the turns of the RPAs.

Ll. 350 – 351: A potentially better sampling strategy has not been discussed. Omit this sentence.

Indeed.  The sentence has been removed.

Technical Corrections

Ll. 2 and 26: Decide on "earth" or "Earth".

Corrected.  We use  "Earth".

L. 14: "maritime" instead of "oceanic".

Corrected.

L. 19: "distribution", not "distributions".

Corrected.

Ll. 133 – 142: The figure uses a slightly different notation for the points in time. (E.g., t=0 and not t0.)

We have homogenized the notation.

L. 150: "RPA" not "RPAS".

Corrected.

Ll. 221 – 225: Where are the panels a to c in Fig. 6?

We have added the panels in the figure.

[Figure]

L. 241: Where is the red arrow in Fig. 7a? And where is Fig. 7a?

We have added the panels and red arrows in the figure.

[Figure]

L. 252: Check the citation style: "Hoffmann et al. (2014)" instead of "(Hoffmann et al. 2014)". Corrected.

L. 271: Add a blank after "profile."

Corrected.

L. 278: There is one parenthesis ")" too many.

Corrected.

---

## Author Comment (AC2)

We wish to thank the reviewer for his/her careful review. Below are our responses (in red) to the comments (in black) on a point-by-point basis. The new text in the manuscript is indicated in quotation marks.

After providing an access review for this article previously, I have now reread it in some more detail. The approach described here is certainly worth pursuing, but I still think the manuscript would greatly benefit from the inclusion of some exploration of the time-evolving case with multiple RPAs. This could be a proof of principle to show that it is possible to characterise the time-evolution of the cloud.

We intend to investigate GPR mapping using a much larger number of RPAs; however, exponentially more computer resources were needed and, unfortunately, we were not able to go beyond two RPAs for this study. Nonetheless, this manuscript shows that two RPAs and GPR mapping are needed to adequately characterize cloud heterogeneities on small enough scales to quantify important parameters such as total LWC.

**We are currently optimizing the simulations to conduct the cloud mapping with several (up to 6) RPAs in a dynamic environment to define temporal scales as a function of the number of RPAs.**

For the time-evolving case, some of the transects in the mature phase (e.g. transect 4 and 7 in figure 15) might resemble the full PDF, but these could be "lucky" transects. Moreover, since the analysis is on single transects here, there does not seem to be an advantage over using single passes with a traditional approach.

We have augmented Fig. 15 with additional simulations to show the robustness of the sampling strategy (that the results are not "lucky").

In principle, many of these exploration techniques can also be applied to traditional aircraft; however the main advantage of using RPA is that multiple platforms can be coordinated simultaneously.

The abstract should at least mention that a single RPA isn't enough to accurately reconstruct individual clouds.

We have updated the abstract to state that a single RPA is not sufficient to accurately reconstruct individual clouds using an adaptive sampling strategy and advanced mapping techniques.

We have added in the abstract :"While a single RPA coupled to GPR mapping remains insufficient to accurately reconstruct individual clouds, two RPAs with GPR mapping adequately characterize cloud heterogeneities on small enough scales to quantify important parameters such as total LWC."

I also think the focus on only 3 clouds (even if these clouds are sampled from multiple starting points) is a weakness of the study. Clouds tend to vary considerably in terms of their shape, especially when they contain multiple updraught cores, so it is hard to see if the results here are generally robust. Showing the LWC convergence for at a few more clouds in the same class size as N2 and N3 would help to establish robustness.

The three clouds used in this study (N1, N2, N3) have not been chosen arbitrarily. We conduct an extensive analysis of the cloud population simulated in the LES to carefully choose the clouds for this study (Figure 4), and show that the three clouds are representative of the cloud population. In addition, they also show similarities with the clouds sampled in Zhao and Austin (2005). Each of the clouds have non-circular crosssections with fractal edges, illustrating the capability of the GPR mapping.

Random sampling of the clouds via different entry points has a similar effect as using multiple clouds by generating unique trajectories that capture the fractal nature of the clouds. Figure 13 establishes the robustness of sampling the different sizes of clouds and the limits of the sampling strategy and length scales.

The length scales for GPR currently seem to be chosen by trial and error, but will depend on both the cloud scale and how well the cloud has been sampled. Note that 75m seems to give a good PDF of LWC, but the LWC RMSE is relatively high. It would also be worth pointing out that clouds are fractal objects, and that this is one of the reasons an ellipse/circle reconstruction fails (another reason is that a transect may not pass through the actual centre).

The length scales are tested for different types of clouds and the value of 75 m remains the most suitable to reconstruct a LWC field, whatever the size of the cloud. And indeed we are well aware of the fractal nature and use Figure 9 to clearly demonstrate that an ellipse/circle reconstruction does not work well. In the discussion of Figure 9, we add in the text that clouds are fractal objects "Clearly, none of these relatively simple methods are able to accurately reconstruct the cloud cross section (Fig.  $\$  ref{volume\_transect}),particularly related to the fractal character of the borders of cloud cumulus."

As mentioned earlier, the target for the center of the clouds is weighted by LWC from GPR mapping and is recalculated after each transect through the cloud.

Regarding the high value of the RMSE for a length scale equal to 75 m (f), the simulated cloud and the reconstructed one were not overlayed properly. We thank the reviewer for catching this. The figure has been rectified as shown below(and has also been updated in the manuscript).

---

## Referee Report (RR1)

**Review of "Use of Large-Eddy simulations to design an adaptive sampling strategy to assess cumulus cloud heterogeneities by Remotely Piloted Aircrafts" by Maury et al. (amt-2021-20)**

The revised version of the manuscript addresses most of my concerns with great care, and I have only very minor and technical comments left. I can now fully support the manuscripts publication in *Atmospheric Measurement Techniques*. I do not need to see the manuscript again.

Please note that line numbers refer to the tracked changes version of the manuscript.

**Minor Revisions**

L. 14: It might be necessary to state that the study is targeted toward "shallow" cumulus clouds.

Ll. 22 – 24: It might better fit the scope of the study if the last sentence of the abstract ends with "[…] on scales small enough to quantify the variability of important parameters such as the LWC."

L. 84: Why did the authors remove the year of the BOMEX campaign?

Ll. 105 – 105: Did the authors use the radiative tendency prescribed in Siebesma et al. (2003), or used a model to determine the longwave cooling rates?

Ll. 236 – 238: While I agree with the statement, how do the authors distinguish between cloud core and cloud edge in Fig. 6?

Ll. 319 – 322: The units for the dissipation rate seem to be incorrect. I assume that a minus is missing in the exponent ($10^{-3}$ instead of $10^3$). The stated value is unrealistically high!

**Technical Corrections**

L. 8: Use the plural for Remotely Piloted Aircrafts here: "Remotely Piloted Aircrafts (RPAs)"

L. 26: "oceans" instead of "ocean regions"

L. 67: "maritime" instead of "marine"

L. 88: "LESs" instead of "Large-Eddy simulations"

L. 93: "LESs" instead of "LES"

L. 95: Remove the period between "m" and "ASL"

L. 116: Add "data" after "high-resolution"

L. 120: The accent aigu is used inconsistently in naming Meso-NH.

L. 269: Add a blank between "and" and "15 %"

L. 274: No comma before "provide"

Ll. 319 ff.: Please rephrase the beginning of this sentence, e.g., "Using equation (5) of Baker et al (1984) allows […]"

**References**

Baker, M. B., Breidenthal, R. E., Choularton, T. W., & Latham, J. (1984). The effects of turbulent mixing in clouds. *Journal of Atmospheric Sciences*, *41*(2), 299-304.

Siebesma, A. P., Bretherton, C. S., Brown, A., Chlond, A., Cuxart, J., Duynkerke, P. G., ... & Stevens, D. E. (2003). A large eddy simulation intercomparison study of shallow cumulus convection. *Journal of the Atmospheric Sciences*, *60*(10), 1201-1219.

---

## Author Response (AR4)

**Review 1 of "Use of Large-Eddy simulations to design an adaptive sampling strategy to assess cumulus cloud heterogeneities by Remotely Piloted Aircrafts" by Maury et al. (amt-2021-20)**

The revised version of the manuscript addresses most of my concerns with great care, and I have

only very minor and technical comments left. I can now fully support the manuscripts publication in

Atmospheric Measurement Techniques. I do not need to see the manuscript again.

Please note that line numbers refer to the tracked changes version of the manuscript.

The authors thank you for your feedback and your acceptance for this article to be published. The authors have taken into account your technical comments

**Minor Revisions**

L. 14: It might be necessary to state that the study is targeted toward "shallow" cumulus clouds.

We have added the term "shallow" in the abstract.

Ll. 22 – 24: It might better fit the scope of the study if the last sentence of the abstract ends with "[...] on scales small enough to quantify the variability of important parameters such as the LWC."

We have replaced the end of the sentence with your proposition.

L. 84: Why did the authors remove the year of the BOMEX campaign? We have added again.

Ll. 105 – 105: Did the authors use the radiative tendency prescribed in Siebesma et al. (2003), or used a model to determine the longwave cooling rates?

We used the prescribed radiative tendency defined in Siebesma et al., 2003. We have added in text a sentence to clarify: "The radiative tendency was prescribed for each hour following the values presented in Siebesma et al., 2003."

Ll. 236 – 238: While I agree with the statement, how do the authors distinguish between cloud core and cloud edge in Fig. 6?

A cross section is observed for each vertical level in the cloud and allows us to determine a core associated with positive velocities (updrafts) and positive buoyancy. The edges are associated with negative velocities (downdrafts) and negative buoyancy. We have added this information as "(studied for each cloud cross section)".

Ll. 319 – 322: The units for the dissipation rate seem to be incorrect. I assume that a minus is missing in the exponent (10 -3 instead of 10 3). The stated value is unrealistically high!

Thank you for catching this mistake. Indeed the minus is missing. We have corrected this value.

**Technical Corrections**

L. 8: Use the plural for Remotely Piloted Aircrafts here: "Remotely Piloted Aircrafts (RPAs)" Corrected. L. 26: "oceans" instead of "ocean regions" Corrected. L. 67: "maritime" instead of "marine" Corrected. L. 88: "LESs" instead of "Large-Eddy simulations" Corrected. L. 93: "LESs" instead of "LES" Corrected. L. 95: Remove the period between "m" and "ASL" Corrected. L. 116: Add "data" after "high-resolution" Corrected. L. 120: The accent aigu is used inconsistently in naming Meso-NH. Corrected. L. 269: Add a blank between "and" and "15 %" Corrected. L. 274: No comma before "provide" Corrected. Ll. 319 ff.: Please rephrase the beginning of this sentence, e.g., "Using equation (5) of Baker et al (1984) allows [...]" Corrected.

**Review 2 of "Use of Large-Eddy simulations to design an adaptive sampling strategy to assess cumulus cloud heterogeneities by Remotely Piloted Aircrafts" by Maury et al. (amt-2021-20)**

Though I think the work would be a worthwhile addition to the literature if some more material is added, I am not yet entirely happy with the draft as it currently stands. The authors have addressed almost all of the minor points I raised, but the larger weaknesses of the study remain. I would encourage the authors to do more work on the draft and resubmit, as I think the topic is important.

We thank the reviewer for his/her careful review. Below are our responses (in red) to the comments (in black) on a point-by-point basis. The text that has changed in the manuscript is indicated in quotation marks.

I still think that the main weakness of the study is that it almost exclusively uses exploration with a single RPA, rather than multiple RPAs. In their reply, the authors acknowledge that for practical applications, the use of multiple RPAs is one of the main advantages over traditional methods. The authors mention that using multiple RPAs exponentially increases computing resources required, but I find it hard to see why these computations would be prohibitively expensive and the cost would grow in a non-linear way, especially for the time-independent case at a single level. I can imagine that a time-dependent GPR with multiple RPAs would require substantial resources, although I think this may be possible provided the algorithm has been coded in an efficiently way. In the abstract, the authors now mention that two RPAs are appropriate, but the draft only demonstrates this for a non-evolving cloud of a certain size class, and it may not work for larger clouds or an evolving cloud.

At the writing of this manuscript, the exploration routines were indeed not efficient and exploration with multiple RPAs in a dynamic environment was prohibitive. We entirely agree with the reviewer that more analysis using multiple RPAs in a dynamic environment is merited -- and is the target for future studies. Nonetheless, the answer to the reviewer's concerns is addressed by averaging individual explorations as there is not (yet) inter-fleet coordination.

Figure 15 extends the exploration to a dynamic case using a single RPA with GPR, which captures the salient features during the cloud's mature phase when its evolution is slower (Figure 15b; 300 and 500 seconds). For the development and dissipation phases (Figure 15b; 50 and 600 seconds) , the multiple explorations on average reproduce the reference PDFs (black line) for LWC between 0.1 and 0.3 g m-3. Averaging the multiple explorations (individual lines in Figure 15b) yields essentially the same results as a dynamic exploration with multiple, uncoordinated RPA. These results show that the 'noise' associated with the reconstructed probability function of LWC is greater during phases with more rapid evolution -- therefore, to reduce the uncertainty, one needs to add RPA. In addition, Figure 15b; 50 and 600 seconds show that the RPA does not capture the higher LWC values associated with the core of the cloud when the cloud element is small, which is a result of a less efficient choice of exploration strategy. In this case, the exploration of small clouds is 'pattern-limited', whereby the Rosette strategy, implemented in Figure 15, is limited by the turn-radius of the RPA relative to the size of the cloud. Other patterns for exploration are currently being studied to address this issue.

To reiterate, we have made significant progress in developing strategies for exploring clouds in this study. In the current manuscript, we show that exploration of a cloud with a single RPA using GPR is far more effective than multiple RPAs without GPR (Figures 11 and 12).

In the text, we have added the following lines in section 3.3.5:

"Averaging the multiple explorations (individual lines in Figure 15b) yields similar results as a dynamic exploration with multiple, uncoordinated RPA. These results show that the 'noise' associated with the reconstructed probability function of LWC is greater during phases with more rapid evolution -- therefore, to reduce the uncertainty, one needs to add RPA. In addition, Figure 15b; 50 and 600 seconds show that the RPA does not capture the higher LWC values associated with the core of the cloud when the cloud element is small, which is a result of a less efficient choice of exploration strategy. "

I am also not convinced that the constant length scale used in the reconstruction is appropriate in every situation. If the cloud would be much larger than N3 in the horizontal dimensions (e.g. a congestus cloud) but would have a similar shape and fractal structure, one would expect the length scale needs to increase as well. There may be reasons to expect that e.g. the scale of the shell does not increase proportionally to cloud scale, but I would still expect that for large clouds, there will at some points be "gaps" when a reconstruction scale of 75m is used. For small cumulus clouds, 75m may be an appropriate choice, but it is good to keep in mind that the scale of 75m is probably close to the effective/actual resolution of the LES (which tends to be a few times larger than the grid spacing) and may also relate to the scale of the gaps between sampled transects.

The sensitivity test to the length scale was first applied on the N2 cloud (equivalent diameter=597 m), which shows that the most efficient length scale is 75 m. The same tests were carried out for the other two clouds, one smaller (Cloud N1, equivalent diameter=240 m) and one larger (Cloud N3, equivalent diameter=1161 m), which resulted in similar length scales averaging 75 +/- 5 m for the three cases (see figure below). The optimum length scale is independent of the cloud size (i.e., N1, N2, N3) suggesting the length scale defined for GPR is related to the length scales of the strongest gradient of the parameter being explored (i.e., LWC in this case). The strongest gradient in LWC occurs in the cloud shell, which is generally two to three grid sizes (i.e., 50 to 75 m) of the LES simulation. The figure below also shows that small length scales (i.e., 25 m) create 'gaps' in the exploration, particularly for large clouds, while large length scales blend the cloud shell and cloud core, which is relatively more important for small clouds.

We have added a paragraph in the manuscript discussing this topic below: "Length scales between 25 m and 400 m were used to find an optimal length scale for Clouds N1, N2, and N3. The sensitivity test was first applied on the Cloud N2 (equivalent diameter=597 m), and the most efficient length scale was found to be 75 m. The same tests were performed for the other two clouds (Cloud N1, equivalent diameter=240 m); Cloud N3, equivalent diameter=1161 m), which resulted in similar length scales, averaging 75 + -5m for the three cases. The optimum length scale is independent of the cloud size (e.g., N1, N2, N3) suggesting the length scale defined for GPR is related to the length scales of the strongest gradient of the parameter being explored (i.e., LWC in this case). The strongest gradient in LWC occurs in the cloud shell, which is generally two to three grid sizes (i.e., 50 to 75 m) of the LES simulation. Length scales that are too small (i.e., 25 m) create 'gaps' in the exploration, particularly for large clouds; while length scales that are too large blend the cloud shell and cloud core, which is relatively more important for small clouds."

The title of section 3.3.4 has also been changed to reflect the analysis: '3.3.4 Exploring clouds of different sizes'

Minor points:

- The authors mention that random sampling of entry points has a similar effect as using multiple clouds. Especially if the clouds have been selected by hand, there is a chance that robustness for some edge cases (like clouds that are broken up at a certain level, or clouds that have a more line-like shape) is not tested for. This may be a small limitation in practice if other information (e.g. radar) is available, but it would be worth mentioning.

We agree and already briefly mention in the conclusion that improvements in a priori information, such as a camera (or as the reviewer has mentioned, a radar) would significantly enhance the cloud exploration. The sentence now reads:

"In order to improve the observational capacity of airborne measurements, various methods are currently being explored, including the use of a camera system or radar to improve the cloud exploration, particularly for conditions when the cloud boundaries are broken or not well defined (e.g., a dissipating cloud)."

- I think the way the GPR works could be described more clearly. In particular, any differences between the way the edge and the interior (cells that fall between different transects) of the cloud are handled are unclear. Note that in Figure 10, there is a large amount of spurious cloud at the outer boundary.

GPR estimates a value and associated uncertainty using the assigned length scales for a given observation (see above response for the length scale analysis). Along the RPA trajectory the values are known, while the prescribed values that extend beyond the RPA trajectory are the results of a weighted ensemble from each of the observed points. The large amounts of spurious clouds result from the extension of the GPR estimates leading to non-zero values outside the cloud -- especially for longer length scales. Note that a small increase in the threshold (to 0.05 g m-3) would eliminate much of the spurious clouds in Figure 10.

- Another minor point is that it is remarkable that two RPAs+GPR eventually give a larger relative error as compared to one (Fig. 12). Do the authors have an explanation for this (it could relate to the previous point)?

The slightly higher relative error at the end of the exploration for two RPAs+GPR with respect to the LWC distribution comes from the spurious points outside the cloud (see response above). These low values of LWC outside the cloud do not impact the total LWC (Figure 11), but do add a small error for low LWC values when comparing to the LWC distribution (Figure 12 and Figure 15). The message from Figure 12 is that two RPAs+GPR yield better results as the cloud is initially being explored -- which is most important, especially in a dynamic environment.

**Minor comments :**

- Some examples of e.g. "the N2 cloud"/"the clouds N1 and N3" (with article) appear in the manuscript.

We have rewritten " Cloud N2 " to be coherent with figures and throughout the manuscript.

- Multiple occurrences: "cloud grids"  $\rightarrow$  "cloudy grid cells" We have replaced 'cloud grids' with 'cloudy grid cells'.

- At places where volumes are calculated, it is still not clear that a 25m thick layer is used (e.g. Figure 9). This needs to be made very explicit (and in fact it would be better to use per area quantities). For the transect method, the 25m grid used to calculate the total LWC also needs to be mentioned explicitly. We have added a statement to specify how the volume is calculated. "The transect method systematically underestimates the cloud volume section (area of the section multiplied by the 25 m thickness of the grid cell)".

Line 53-55: "These campaigns"/"These LES" needs to be expanded (as "these" does not refer back to the previous sentence).
We have replaced 'these' with "the ".

- Line 74: "in static case"  $\rightarrow$  "in a static case" Corrected

- Line 75: "field"  $\rightarrow$  "fields" Corrected

- Line 101: "is not"  $\rightarrow$  "are not"... "a two-moment microphysics scheme" (article and plural needed, also in other places) We replaced 'is' with 'are' and "a two-moment microphysic scheme" by a twomoment microphysical scheme".

- Line 109 etc.: meshes  $\rightarrow$  grid points Corrected

Line 121: replace second "of" by e.g. "who argued" Corrected
Line 125: change word order "varies between..." We have changed the word order.

- Line 135; "for the first time"  $\rightarrow$  not sure what is meant here. We have deleted this term.

- Line 155: a RPA airspeed  $\rightarrow$  an RPA airspeed (similarly "an RPA in line 159 and 322) Corrected

- Line 258: " PDF distribution"  $\rightarrow$  "PDF", "and 15"  $\rightarrow$  add space Corrected

- Line 259: equal to  $\rightarrow$  between X ...and Y. (indicate range, rather than single value) Corrected. "between 0.7 to 0.9 m.s-1."

- Line 270: a space is missing in this line. Corrected

- Line 306: reword to "the formula of Baker et al. "  $\rightarrow$  This needs explanation: it would be good to repeat the formula of Baker et al. We have reformulated by: "Using equation (5) of Baker et al. (1984) allows relating the time needed for".

- Line 308: "allow us to" Corrected.

- Line 310: "in the hypothesis of a dynamic cloud"  $\rightarrow$  I am not sure what is meant here.

We have replaced this phrase by "in the case of a dynamic cloud."

- Line 334: "pattern-limited": explain in text.

We have added: "in that the RPA cannot turn around and re-enter the cloud if the cloud itself is smaller than the RPA's turning radius. "

- Line 350: note how the reference frame might be derived in practical applications.

We have added "which is redefined at each time step by accounting for the advective wind."

```
Line 374: "intercomparison" (for consistency)
Corrected.
Line 375 "high-frequency"
Corrected
Line 403: "allows us to"
Corrected
Figure 8 caption: replace "alt= 700 m" by "a height of 700 m ASL".
Corrected
```